# FLEX3D: FEED-FORWARD 3D GENERATION WITH FLEXIBLE RECONSTRUCTION MODEL AND INPUT VIEW CURATION

## ABSTRACT

Generating high-quality 3D content from text, single images, or sparse view images remains a challenging task with broad applications. Existing methods typically employ multi-view diffusion models to synthesize multi-view images, followed by a feed-forward process for 3D reconstruction. However, these approaches are often constrained by a small and fixed number of input views, limiting their ability to capture diverse viewpoints and, even worse, leading to suboptimal generation results if the synthesized views are of poor quality. To address these limitations, we propose Flex3D, a novel two-stage framework capable of leveraging an arbitrary number of high-quality input views. The first stage consists of a candidate view generation and curation pipeline. We employ a fine-tuned multi-view image diffusion model and a video diffusion model to generate a pool of candidate views, enabling a rich representation of the target 3D object. Subsequently, a view selection pipeline filters these views based on quality and consistency, ensuring that only the high-quality and reliable views are used for reconstruction. In the second stage, the curated views are fed into a Flexible Reconstruction Model (FlexRM), built upon a transformer architecture that can effectively process an arbitrary number of inputs. FlexRM directly outputs 3D Gaussian points leveraging a tri-plane representation, enabling efficient and detailed 3D generation. Through extensive exploration of design and training strategies, we optimize FlexRM to achieve superior performance in both reconstruction and generation tasks. Our results demonstrate that Flex3D achieves state-of-the-art performance, with a user study winning rate of over 92% in 3D generation tasks when compared to several of the latest feed-forward 3D generative models.

See `anonymous project page` for more immersive 3D results.

## 1 INTRODUCTION

Fast generation of high-quality 3D contents is becoming increasingly important for video games development (Hao et al., 2021; Sun et al., 2023), augmented, virtual, and mixed reality (Li et al., 2022), robotics (Nasiriany et al., 2024) and many other applications. Recent advances in computer vision and graphics (Mildenhall et al., 2021; Kerbl et al., 2023) and deep learning (Dosovitskiy, 2020; Caron et al., 2021; Oquab et al., 2023), combined with the availability of large datasets of 3D objects (Deitke et al., 2023; 2024; Yu et al., 2023; Chang et al., 2015), have made it possible to learn neural networks that can generate 3D objects from text, single images or a sparse set of views, and to do so in an feed-forward manner, achieving significantly faster speeds than distillation-based methods (Poole et al., 2022; Li et al., 2023; Qiu et al., 2024; Chen et al., 2023; Wang et al., 2023b).

A particularly successful family of 3D generators are the ones based on sparse-views reconstruction (Xu et al., 2024c; Siddiqui et al., 2024; Li et al., 2024b; Zhang et al., 2024c; Tang et al., 2024a; Xu et al., 2024b; Xie et al., 2024a; Wang et al., 2024c). Compared to single-image reconstructors, multi-view reconstruction models generally produce better 3D assets. This advantage arises because the multi-view images implicitly capture the object geometry much better, substantially simplifying the reconstruction problem. However, to generate a 3D object from text or a single image, one must first synthesize several views of the objects, for example by means of a multi-view diffusion model. These multi-view diffusion models often generate inaccurate and inconsistent views, which are dif-

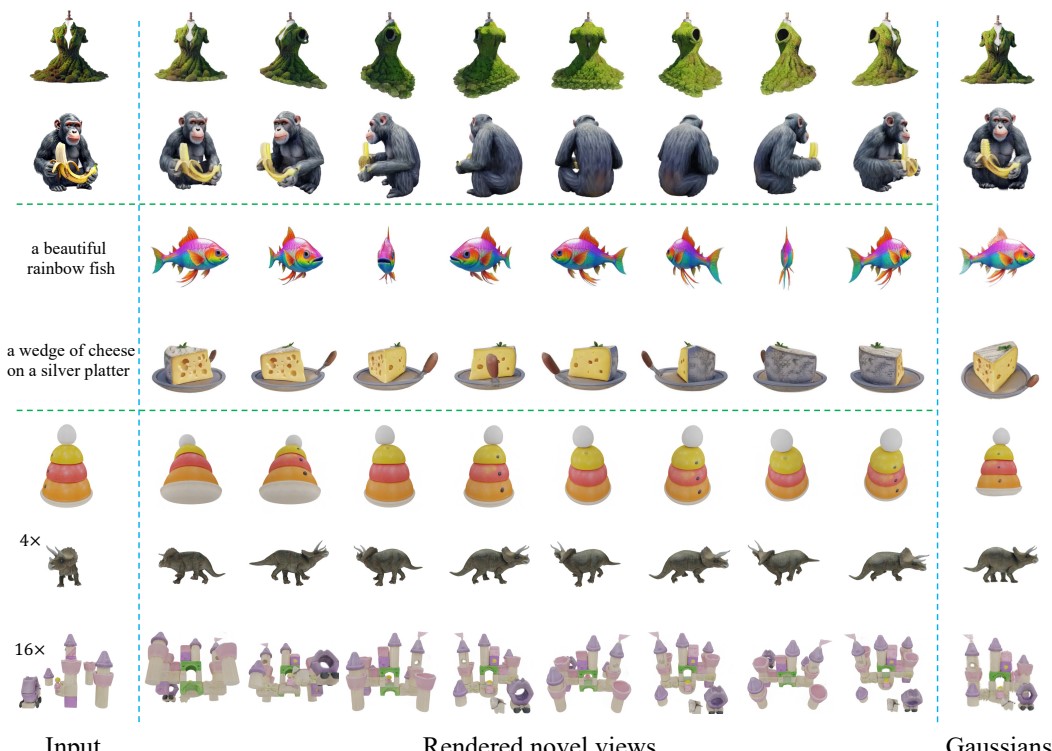

Figure 1: **Results produced by Flex3D**. It generates high-quality 3D Gaussians from a single image, textual prompt, and performs 3D reconstruction from an arbitrary number of input views.

ficult for the reconstruction network to reconcile, and can thus affect the overall quality of the final 3D output (Tang et al., 2024c).

This paper thus focuses on the problem of generating a high-quality set of different views of an object, with the goal of improving the quality of the final 3D object reconstruction. We build on a simple observation: the quality of the 3D reconstruction improves as the quality and quantity of the input views increases (Han et al., 2024b). Hence, instead of relying on a fixed, limited set of views generated by potentially unreliable multi-view diffusion models, we suggest to generate a pool of candidate views and then automatically select the best ones to use for reconstruction.

Based on this idea, we introduce a new framework, *Flex3D*, comprising a new multi-view generation strategy as well as a new flexible feed-forward reconstruction model.

First, we propose a mechanism to generate a large and diverse set of views. We do this by training two diffusion models, one that generates novel views at different azimuth angles and the other at different elevation angles. The models are designed to make the views as consistent as possible. Second, we propose a view selection process that uses a generation quality classifier and a feature matching network to measure the consistency of the different views. The result of this selection is a good number of high-quality views, which help to improve the quality of the final 3D reconstruction.

Differently from many prior works, then, we need to reconstruct the 3D object from a variable number of views which depends on what the selection process returns. Hence, we require a reconstruction model that (1) can ingest a varying numbers of input views and different viewing angles; (2) is memory and speed efficiency to handle a large number of input views; and (3) can output a full, high-quality 3D reconstruction of the object, regardless of the number and pose of input views.

To this end, we introduce *Flexible Reconstruction Model* (FlexRM). FlexRM starts from the established Instant3D architecture (Li et al., 2024b) and adds a stronger camera conditioning mechanism to address the first requirement (1). It also introduces a simple but effective way of combining the Instant3D tri-plane representation with 3D Gaussian Splatting, meeting requirements (2) and (3).

Specifically, FlexRM learns a Multi-Layer Perceptron (MLP) to decode the tri-plane features into the parameters of 3D Gaussians used to represent the object. We also simplify the process of learning this MLP, thus leading to notable performance improvements, by pre-training parts of it, where we initialize the color and opacity parts using an off-the-shelf NeRF (Mildenhall et al., 2021) MLP. For the remaining Gaussian parameters, we learn rotation and scale in a conventional manner while learning position offsets which are combined with the tri-plane feature sampling locations.

While our view selection pipeline identifies the best views for reconstruction, it still does not eliminate all multi-view inconsistencies. To mitigate the impact of the minor inaccuracies that remain, FlexRM employs a novel training strategy. Although our training dataset consists of perfectly rendered images, we simulate imperfections in the input views by leveraging the output of FlexRM itself. Specifically, we take FlexRM's reconstructed 3D Gaussians, add noise to them, and generate new noisy views of the object based on these noisy Gaussians. Compared to directly manipulating the views, this approach allows us to inject more expressive and representative types of noise, as shown in fig. 3. The noisy views are then combined with clean rendered views and fed as input to the 3D reconstructor, with the goal of producing clean, noise-free representations of the 3D object. This approach enables the model to learn how to handle imperfect inputs.

We benchmark our method against state-of-the-art feed-forward models in 3D generation and 3D reconstruction tasks, evaluating performance through a user study and various automated metrics. We achieve the best results in reconstruction tasks across all settings (single-view, four-view, and more-view), as well as in generation tasks. We also conduct a thorough ablation study to assess the impact of our design choices.

In summary, this paper makes the following key contributions to address the limitations of current two-stage 3D generation pipelines: (1) We introduce a novel pipeline that generates a pool of 2D views of an object and only selects the optimal subset for 3D reconstruction. (2) We propose FlexRM, a 3D reconstruction network that efficiently processes an arbitrary number of input views with varying viewpoints, enabling high-quality feed-forward reconstruction from the selected views. (3) We introduce a novel training strategy to enhance the robustness of the 3D reconstructor by simulating imperfect input views. This improves FlexRM's resilience to small noise in the input data that may remain.

## 2 RELATED WORK

### 2.1 MULTI-VIEW GENERATION

Generating novel views from a single image or text without learning a 3D representation is a highly ill-posed and challenging task. With the development of image and video diffusion models, this task has become easier to address, as solutions can be built upon these pre-trained models.

Zero123 (Liu et al., 2023a) first proposed using multi-view data to fine-tune an image diffusion model for generating novel views from a single view, conditioned on camera parameters. Following this approach, subsequent works (Li et al., 2024b; Shi et al., 2023b; Tang et al., 2023; Liu et al., 2023b; Long et al., 2024; Wang & Shi, 2023; Woo et al., 2024; Yang et al., 2024b; Ye et al., 2024; Zhao et al., 2024; Zheng & Vedaldi, 2024; Tang et al., 2024b) largely focused on generating multiple views simultaneously to ensure 3D consistency.

With the availability of powerful video diffusion models, recent works (Kwak et al., 2023; Voleti et al., 2024; Melas-Kyriazi et al., 2024; Li et al., 2024a; Han et al., 2024a; Gao et al., 2024; Zuo et al., 2024; Yang et al., 2024a) have adopted them to improve multi-view generation. However, none of these models can reliably generate a large number of perfectly consistent views. Furthermore, even with camera conditioning, models like SV3D (Voleti et al., 2024) perform poorly when the specified elevation angle deviates from zero. This justify our approach of selecting the best views from a pool of generated views.

### 2.2 FEED-FORWARD 3D RECONSTRUCTION AND GENERATION

Recent advances in 3D reconstruction and generation have focused on training feed-forward models that directly output 3D representations without requiring further optimization (Yu et al., 2021; Erkoç

Figure 2: **Flex3D** comprises two stages: (1) candidate view generation and selection, and (2) 3D reconstruction using FlexRM. In the first stage, an input image or textual prompt drives the generation of a diverse set of candidate views through fine-tuned multi-view and video diffusion models. These views are subsequently filtered based on quality and consistency using a view selection mechanism. The second stage leverages the selected high-quality views, feeding them to FlexRM which reconstruct the 3D object using a tri-plane representation decoded into 3D Gaussians.

et al., 2023; Szymanowicz et al., 2023b; Ren et al., 2023; Lorraine et al., 2023; Xu et al., 2024a; Tochilkin et al., 2024; Zhang et al., 2024a; Jiang et al., 2024; Han et al., 2024a). These feed-forward models offer significant advantages in both reconstruction quality and inference speed.

A representative series of work is LRM (Hong et al., 2024), which learns to generate a tri-plane NeRF (Chan et al., 2022) representation using a transformer network. This approach can receive multiple types of inputs, including single images, text (Xu et al., 2024d), posed sparse-view images (Li et al., 2024b), and unposed sparse-view images (Wang et al., 2024a). Further works (Wei et al., 2024; Siddiqui et al., 2024; Xu et al., 2024b; Wang et al., 2024c; Boss et al., 2024) focused on improving the geometric quality of generated 3D assets. Some (Zou et al., 2023) proposed to combine the tri-plane representation with 3D Gaussian Splatting for more efficient rendering. They suggest using an additional point cloud network to determine the 3D Gaussian position to overcome the tendency of 3D Gaussian to get stuck in local optima.

Another representative series of work (Tang et al., 2024a; Xu et al., 2024c; Zhang et al., 2024c) generates 3D Gaussian points directly through per-pixel aligned Plücker ray embedding and predicts the depth for each pixel (Szymanowicz et al., 2023a), which can then be converted to 3D Gaussian locations. However, such approaches requires the input views to cover a large visible range of the 3D object. Building an intermediate 3D feature representation to regress 3D Gaussian points is also possible (Chen et al., 2024a; Zhang et al., 2024b), but these methods still require sparse-view images as input and typically prefer a fixed number of views with fixed viewing angles. However, in 3D generation tasks where sparse input views are generated through a multi-view diffusion model and are not always of high quality, such sparse-view reconstructors tend to produce suboptimal results.

This paper introduces a flexible 3D reconstruction model that combines the strengths of the approaches above. Our tri-plane-based model efficiently generates high-quality 3D Gaussian points directly, without needing additional modules. It also accommodates a variable number of input views, enabling integration with our view selection pipeline for high-quality 3D generation.

## 3 METHOD

We illustrate our method in fig. 2. We begin by presenting our approach for generating a pool of candidate views and the subsequent selection process in section 3.1. We then describe the design of the FlexRM in section 3.2. Finally, we outline our training strategy that simulates imperfect input views in section 3.3.

### 3.1 CANDIDATE VIEW GENERATION AND SELECTION

Here we describe how a pool of candidate views is generated from a single image or text and then filtered for quality and consistency before performing 3D reconstruction.

**Multi-view generation at varying elevations.** Our image/text-to-multi-view-images generator module generates four views of the 3D object from four elevation degrees (-18°, 6°, 18°, and 30°) We utilize the Emu model (Dai et al., 2023), which is pre-trained on a massive dataset of billions of text-image pairs, as our base model. Following prior works (Shi et al., 2023b; Li et al., 2024b; Siddiqui et al., 2024), we fine-tune this model on approximately 130,000 rendered multi-view images. This fine-tuning process enables the model to predict a $2\times2$ grid of four consistent images, each corresponding to one of the specified elevation angles.

**Multi-view generation at varying azimuths.** After generating four views at varying elevations, we employ a fine-tuned Emu video model (Girdhar et al., 2023; Melas-Kyriazi et al., 2024; Han et al., 2024a) to generate a video with 16 views spanning a full 360° azimuth range. This model is fine-tuned on a dataset encompassing a wide spectrum of elevation angles, enabling it to generate consistent, high-quality views from diverse inputs with varying elevations. We generate the multi-view videos starting from the input view at 6° elevation, which usually results in representative views for the subsequent reconstruction process.

**View selection.** As the two multi-view diffusion models are focused on different aspects (elevation and azimuth) of novel view generation, there are minimal conflicts in the generated views between them. Even so, and despite efforts to improve the quality of the outputs, not all generated views are entirely consistent. Certain views, particularly those from challenging angles like the back, may exhibit suboptimal generation quality, and there can be inconsistencies between different generated views. Including such flawed views as input for 3D reconstruction can significantly degrade the quality of the final 3D asset. Therefore, we introduce a mechanism to filtering poor-quality views. This is done via a novel view selection pipeline which consists of two steps:

(1) *Back View Quality Assessment:* We employ a multi-view video quality classifier trained to assess the overall quality of generated videos, with particular emphasis on the quality of the back view. This classifier utilized DINO (Oquab et al., 2023) to extract features from the front view and back view of the multi-view video, and subsequently trained a Support Vector Machine (SVM) to classify video quality based on the combined DINO features. The training data consisted of 2000 manually labeled "good" and "bad" Emu-generated video samples.

We apply the quality classifier to the multi-view video to determine whether the back view exhibits reasonable generation quality.

(2) *Multi-View Consistency Verification:* If the back view quality is deemed acceptable, we designate both the back and front views as initial query views. The front view typically possesses the highest visual quality, as it is directly based on the input provided to the fine-tuned EMU video diffusion model. Conversely, if the back view quality is inadequate, only the front view serves as the initial query view. We utilize the Efficient LoFTR (Wang et al., 2024b) to match features between all 20 generated views and the selected query views. Views with matching point counts exceeding the mean minus 60% of the standard deviation are added to the selected results. This step effectively gathers high-quality side views and views at different poses that demonstrate strong consistency with the initial query views.

Candidate view generation typically takes about a minute on a single H100 GPU, and the whole process of view selection can be done in less than a second with a single A100 GPU.

## 3.2 FLEXIBLE RECONSTRUCTION MODEL (FLEXRM)

As outlined in the introduction, FlexRM aims to fulfill the following requirements: (1) adaptability to varying numbers of input views and their corresponding viewing angles, (2) memory and speed efficiency, and (3) the ability to infer a full 3D reconstruction of the object independently of how many and which views are available. We follow a minimalist design philosophy and strive to minimize modifications to Instant3D, enabling easy reuse of weights from architectures like Instant3D, and simplifying implementation.

**Stronger camera conditioning.** Handling varying numbers of input views with viewing angles necessitates providing camera information to the network. In Instant3D, each view's camera information, including its extrinsic and intrinsic parameters, is represented as a 20-dimensional vector. This vector is then passed through a camera embedder to produce usually a 1024-dimensional camera feature, which is subsequently injected into the DINO (Oquab et al., 2023) image encoder net-

work (responsible for extracting image features for each view) using an AdaIN (Huang & Belongie, 2017) block. The image encoder's final output comprises a set of pose-aware image tokens, which has 1024 768-dimensional tokens for every $512 \times 512$ resolution input view. These per-view tokens are concatenated to form the feature descriptors for input views.

Our aim is to ensure that the DINO-extracted tokens are not only camera-aware, but also explicitly incorporate learnable camera information into the final feature descriptors. To achieve this, we set the output dimension of the camera embedder to 768, enabling it to match the dimension of the pose-aware image tokens and be appended to them, resulting in 1025 (1024 image tokens + 1 camera token) 768-dimensional tokens overall. This simple way of attaching explicit camera information enhances the network's camera awareness, strengthening its performance in complex scenarios where a large number of input views is provided.

**Bridging tri-planes and 3D Gaussian Splatting.** For rendering speed and memory efficiency, we opt to use 3D Gaussian Splatting (3DGS) (Kerbl et al., 2023) to represent the 3D object. However, Instant3D uses a tri-plane NeRF representation instead. To bridge these two, we predict a set of 3D Gaussian from the tri-plane features via an MLP. Because 3DGS is notoriously sensitive to the initial Gaussian parameters, we carefully initialize both the MLP predictor and the tri-plane transformer network with an off-the-shelf tri-plane NeRF network.

A tri-plane is a compact representation of a volumetric function $[-1, 1]^3 \rightarrow \mathbb{R}^d$ mapping points $\mathbf{p} \in [-1, 1]^3$ to feature vectors $f(\mathbf{p}) \in \mathbb{R}^d$. Starting from an initial position $\mathbf{p}_0$, the model first reads off the corresponding tri-plane feature $f(\mathbf{p}_0)$, and then feeds the latter into an MLP to predict the parameters of a corresponding 3D Gaussian, namely, its position, color, opacity, rotation, and scaling. To obtain a mixture of such Gaussians, we simply start from a set of initial positions $\mathbf{p}_0$ and apply the MLP at each location. We use a $100 \times 100 \times 100$ grid to sample the initial positions, resulting in the prediction of 1 million Gaussians.

The position of the 3D Gaussian is expressed as $\mathbf{p} = \alpha \mathbf{p}_0 + (1 - \alpha) \delta \mathbf{p} = \alpha \mathbf{p}_0 + (1 - \alpha) \operatorname{MLP}(f(\mathbf{p}_0))$ where $\delta \mathbf{p}$ is a a positional offset output by the MLP and $\alpha = 3/4$. These positional offsets $\delta \mathbf{p}$ are constrained to the range of $[-1, 1]$ through the application of the tanh activation function. This approach, akin to residual learning, facilitates the optimization process. The multipliers ensure that $\mathbf{p}$ remains within the same range as $\mathbf{p}_0$, which prevents Gaussian points from moving beyond the visible boundaries, which would result in their projection falling outside the 2D image plane and consequently providing no gradients for optimization. Furthermore, since $\alpha < 1$, this expression biases Gaussians to shift towards the center of the tri-plane grid, where the object is usually located.

The color and opacity of the Gaussian are output by the same MLP that, in Instant3D, outputs the color and opacity of their NeRF representation. These two parameters need no conversion as color and opacity in NeRF and 3DGS are similar in functionality. Finally, the part of the MLP that predicts the Gaussian rotation and scaling parameters are learned from scratch.

**Data.** Our training dataset comprises multi-view dense renders from an internal dataset analogous to Objaverse. Specifically, for each object, we render $512 \times 512$ resolution images from 256 viewpoints, uniformly distributed across 16 azimuth and 16 elevation angles. This process yields approximately 700,000 rendered objects, with 140,000 classified as high-quality. Furthermore, we leverage the Emu-video synthetic dataset (Han et al., 2024a), which consists of 2.7 million synthetic multi-view videos. Each video comprises 16 frames capturing a 360-degree azimuth at a fixed elevation angle.

**Two-stage training.** Initially, we pre-train FlexRM using a NeRF MLP architecture. This stage employs the Emu-video synthetic data, where we randomly select 1 to 16 input images ($256 \times 256$ resolution) and render 4 views with fixed rendering resolution of $256 \times 256$ and patch resolution of $128 \times 128$ for supervision (L2, LPIPS (Zhang et al., 2018), and opacity). The pre-training phase aims to provide a good initialization for the subsequent GS MLP training and is conducted for 10 epochs, requiring 2 days on 64 A100 GPUs.

For the second GS training stage, we utilize the 700,000 dense renders. A random number (between 1 and 32) of input images ($512 \times 512$ resolution) are fed into FlexRM, and we render 4 novel views ($512 \times 512$ resolution) to compute losses. Given that our dense renders encompass images with diverse elevation angles, we implement weighted sampling for both input and rendered images. This assigns higher selection probabilities to images with elevation angles closer to zero. The training process spans 20 epochs and takes 4 days on 128 A100 GPUs.

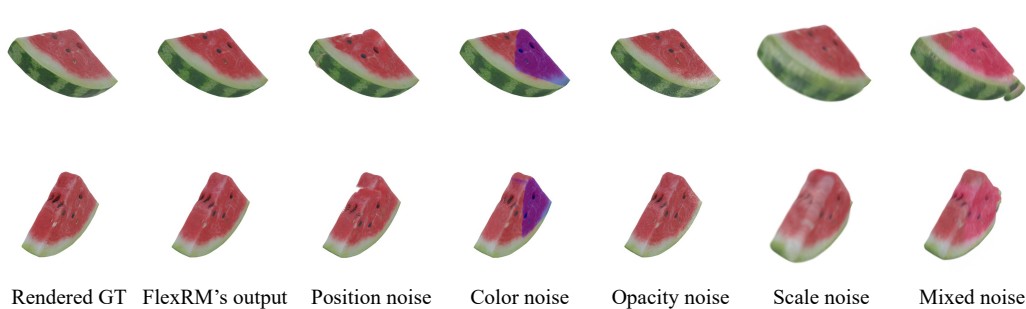

Rendered GT  FlexRM's output  Position noise  Color noise  Opacity noise  Scale noise  Mixed noise

Figure 3: **Imperfect Input View Simulation Results**. We simulate different kinds of imperfect input views by feeding FlexRM's output back as input and manipulating 3D Gaussian parameters.

FlexRM generates 1M 3DGS points in under 0.5 second and renders in real time with a single A100 GPU. Increasing the number of input images has only slight impact on speed and memory usage.

More details on implementation and training of FlexRM are presented in the Appendix A.

### 3.3 IMPERFECT INPUT VIEW SIMULATION

Even after input view selection, these views may still contain minor imperfections. To enhance FlexRM's suitability for generation tasks, we require it to be robust to such imperfections while maintaining high-quality 3D outputs. We achieve this robustness by simulating imperfect inputs during a fine-tuning stage.

This necessitates simulating a wide variety of imperfections efficiently. Simply adding noise in image space makes it difficult to simulate imperfections arising from geometric distortions. Instead, we propose a three-step process: (1) First, we perform inference using FlexRM with a small random number of rendered images (between 1 and 8) as inputs to generate another random number of images (between 1 and 32) for subsequent use as inputs. (2) Next, we use these generated images to replace the rendered images at the same viewing angles with a 50% probability, forming a new input set. This replacement probability is based on the observation that multi-view diffusion-generated images typically exhibit inconsistencies and imperfections non-uniformly. Additionally, this approach encourages the reconstructor to focus more on high-quality input views. (3) Finally, we re-run FlexRM with gradients enabled, using the new input set as inputs and supervising it with novel view rendering losses, while utilizing perfectly rendered views as ground truth. This fine-tuning process enables FlexRM to learn to tolerate minor imperfections in input views and still produce high-quality 3D reconstructions.

While FlexRM's outputs naturally contain small imperfections, we also introduce random perturbations to FlexRM's generated 3D Gaussian points during step (1) to simulate a wider range of imperfect inputs and promote greater diversity. We sample a randomly sized small cube from the large tri-plane cube and add noise with varying intensities to the 3D Gaussian parameters, excluding rotation. For example, adding noise to positions can simulate part-level 3D inconsistencies, while adding noise to color parameters can simulate color distortions. Adding noise to opacity results in a speckled or streaky appearance, while adding noise to scale leads to a blurring effect. Figure 3 illustrates these effects. During training, each effect is randomly used with a probability of 0.2. This combines them for greater diversity. Please check Appendix A for more details.

## 4 EXPERIMENTS

We evaluate Flex3D on the 3D generation (section 4.1) and 3D reconstruction (section 4.2) tasks, comparing it to state-of-the-art methods and ablating various design choices (section 4.3).

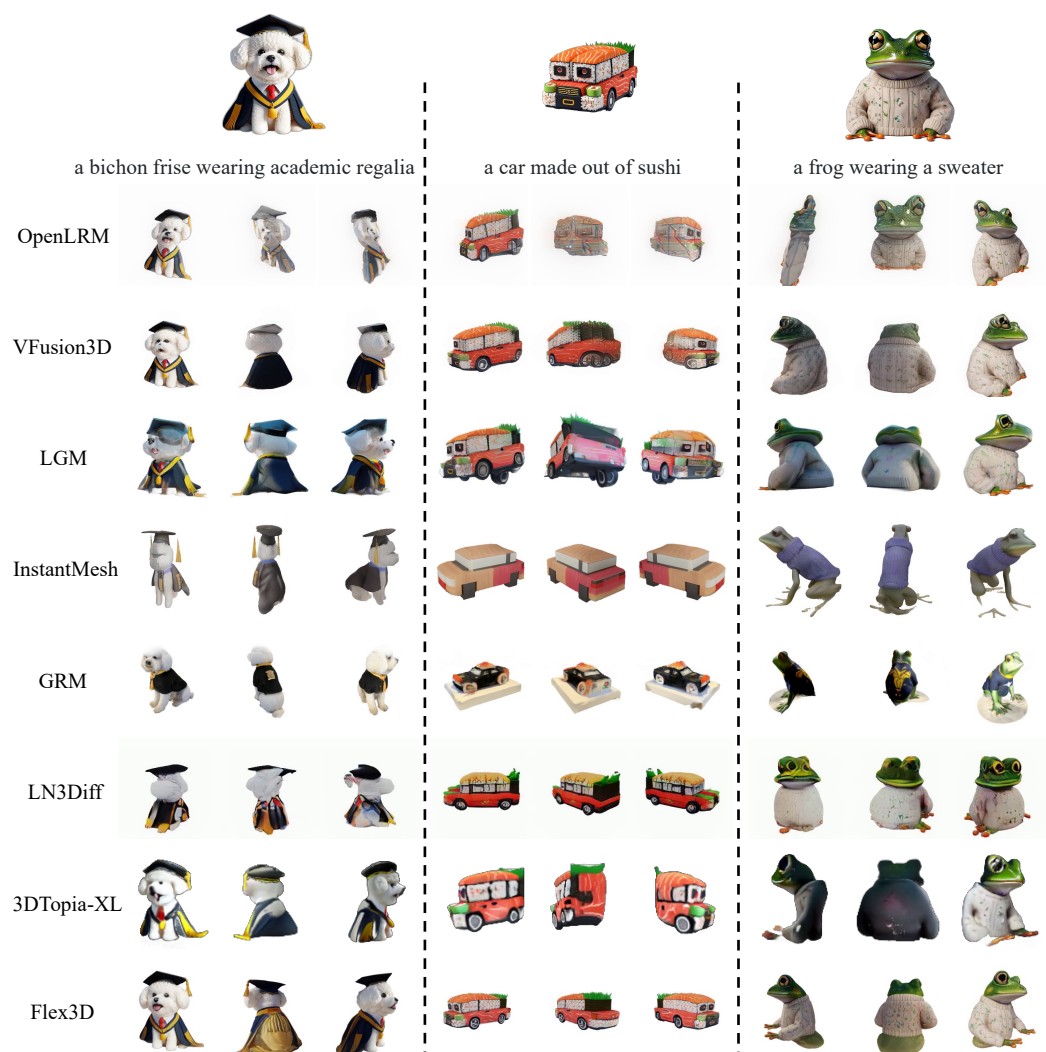

Figure 4: **Qualitative Results of Text-to-3D Generation**. Flex3D demonstrates higher generation quality with strong 3D consistency, outperforming all other methods.

| Method | CLIP text similarity↑ | VideoCLIP text similarity↑ | Flex3D win rate |
|---|---|---|---|
| OpenLRM | 0.243 | 0.229 | 100 % |
| VFusion3D | 0.265 | 0.238 | 95.0 % |
| LGM | 0.266 | 0.240 | 97.5 % |
| InstantMesh | 0.272 | 0.236 | 95.0 % |
| GRM | 0.268 | 0.253 | 92.5 % |
| LN3Diff | 0.252 | 0.234 | 95.0 % |
| 3DTopia-XL | 0.254 | 0.231 | 97.5 % |
| Flex3D | **0.277** | **0.255** | - |

Table 1: **Comparisons on 3D Generation Task.** Flex3D achieves the highest scores for both CLIP text similarity and VideoCLIP text similarity, exhibiting better performance in text alignment. For generation quality, we conduct a user study to assess it, and the winning rate of Flex3D is always greater than 92%, demonstrating its strong generation performance.

## 4.1 3D GENERATION

We leverage 404 deduplicated prompts from DreamFusion (Poole et al., 2022) to conduct an experiment on text-to-3D or single-image-to-3D generation. We compare Flex3D to a few recent

feed-forward 3D generation methods including OpenLRM (He & Wang, 2023; Hong et al., 2024), VFusion3D (Han et al., 2024a), LGM (Tang et al., 2024a), InstantMesh (Xu et al., 2024b), and GRM (Xu et al., 2024c). We also compare Flex3D with two recent direct 3D generation (diffusion) methods: LN3Diff (Lan et al., 2024) and 3DTopia-XL (Chen et al., 2024b). For GRM, we utilize its provided Instant3D's multi-view diffusion model to generate input multi-view images, and for InstantMesh, we employ the default Zero123++ (Shi et al., 2023a) for text-to-input multi-view image conversion. For direct 3D diffusion methods, we use their single-image-to-3D generation pipeline.

We present qualitative results in fig. 4. Our model demonstrates strong generation capabilities with good global 3D consistency and detailed high-quality textures. Quantitative results are presented in table 1, where Flex3D outperforms all baselines, showing high alignment in text prompt and generated content.

To further evaluate the overall quality of the generated content, we conducted a user study. Participants were presented with pairs of 360° rendered videos—one generated by Flex3D and one by a baseline model—and asked to select their preferred video. We randomly selected 40 prompts for evaluation. The corresponding 40 pairs of generated videos were then independently evaluated by five users, with each user assessing all 40 pairs. For each pair, we collect five results, and the majority preference was recorded as a win rate for Flex3D. Results are also shown in table 1, where a significant number of votes goes to our method for its high quality, regardless of the baselines used for comparison. This shows that our method clearly generates better 3D assets.

## 4.2 3D RECONSTRUCTION

We utilize the Google Scanned Objects (GSO) dataset (Downs et al., 2022) for evaluation. From this dataset, we use 947 objects excluding some shoes that are so similar to be redundant. Each object is rendered at $512 \times 512$ resolution from 64 different viewpoints, which are generated using four elevation settings: -30°, 10°, 30°, and 45°. The azimuth angles are uniformly sampled between 0° and 360°.

| Method | Input views | PSNR↑ | SSIM↑ | LPIPS↓ | CLIP image sim↑ | CD↓ | NC↑ |
|---|---|---|---|---|---|---|---|
| OpenLRM | 1 | 15.83 | 0.821 | 0.209 | 0.602 | - | - |
| VFusion3D | 1 | 19.10 | 0.827 | 0.158 | 0.759 | - | - |
| FlexRM | 1 | **21.21** | **0.862** | **0.125** | **0.832** | - | - |
| InstantMesh | 4 | 21.33 | 0.859 | 0.133 | 0.809 | 1.372 | 0.841 |
| GRM | 4 | 25.03 | **0.899** | 0.102 | 0.869 | 1.496 | 0.866 |
| FlexRM | 4 | **25.55** | 0.894 | **0.074** | **0.893** | **1.205** | **0.878** |
| FlexRM | 8 | 26.33 | 0.897 | 0.069 | 0.906 | 1.188 | 0.881 |
| FlexRM | 16 | 26.51 | 0.902 | 0.068 | 0.911 | 1.182 | 0.884 |
| FlexRM | 24 | 26.65 | 0.905 | 0.067 | 0.915 | 1.175 | 0.886 |
| FlexRM | 32 | 26.77 | 0.907 | 0.066 | 0.919 | 1.169 | 0.888 |

Table 2: **Reconstruction Performance on the GSO Dataset.** FlexRM consistently outperforms other baselines, where it achieves the best results across different input view settings. CD (Chamfer Distance) values are multiplied by $10^{-2}$, and NC denotes Normal Correctness. Increasing the number of input views for FlexRM leads to improved reconstruction quality.

In table 2 we report the performance of FlexRM on the GSO reconstruction task, using varying numbers of input views, namely 1, 4, 8, and 16. We compare our results to several baseline methods, including single-view reconstruction models (LRM (He & Wang, 2023), VFusion3D (Han et al., 2024a)) and sparse-view reconstruction models (InstantMesh (Xu et al., 2024b), GRM (Xu et al., 2024c)). For the single-view setting, we use the input view at 0° azimuth and 10° elevation as input. For the 4-view setting, we use views at 0°, 90°, 270°, and 360° azimuth degrees, all at 10° elevation. For other numbers of views, we heuristically select more views as input. The remaining views are used to compute the reported novel-view synthesis quality. The CD (Chamfer Distance) and NC (Normal Correctness) calculation protocol follows that of AssetGen (Siddiqui et al., 2024).

Overall, FlexRM significantly outperforms baselines in both 1-view and 4-view settings. This improvement is particularly evident in the LPIPS score, a key metric reflecting perceptual quality, which demonstrates substantial gains. Beyond fixed input views, FlexRM is also capable of han-

dling an arbitrary number of input views. We present results for more view results, both exhibiting progressively stronger performance. Qualitative results are shown in the Appendix D.

### 4.3 ABLATION STUDY AND ANALYSIS

**FlexRM in 3D Reconstruction.** We first ablate various design choices of FlexRM using 3D reconstruction metrics, including: (1) not using the stronger camera conditioning, (2) directly predicting positions ($\alpha = 1$), (3) not using positional offsets ($\alpha = 0$), and (4) not using two-stage training. All experiments here are conducted on a weaker version of FlexRM, trained with 140,000 data points and for 20 epochs only in stage 2. We use the same evaluation setting as in section 4.2.

Results are shown in table 3, where we report the averaged results across four different settings: 1, 4, 8, and 16 input views. Overall, removing each component leads to a performance drop. Notably, directly predicting positions and removing positional offsets result in significant performance decreases. This highlights the importance of accurately modeling Gaussian positions for high-quality reconstruction. Interestingly, removing stronger camera conditioning has a relatively smaller impact on performance. This is because the advantages of stronger camera conditioning become more pronounced when a larger number of input views with varying camera poses are used. To validate this, we also test a 32-view input experiment, where incorporating stronger camera conditioning improved PSNR by over 0.3 dB.

| Ablation | PSNR↑ | SSIM↑ | LPIPS↓ | CLIP image sim↑ |
|---|---|---|---|---|
| No stronger camera cond | 24.31 | 0.871 | 0.092 | 0.865 |
| Directly predict positions | 23.41 | 0.840 | 0.096 | 0.831 |
| No positional offsets | 22.19 | 0.798 | 0.102 | 0.789 |
| No two-stage training | 23.38 | 0.838 | 0.098 | 0.827 |
| Full model | **24.35** | **0.873** | **0.090** | **0.868** |

Table 3: **Ablation Study of FlexRM.** We evaluate the impact of removing individual components of our proposed method.

**Flex3D in 3D Generation.** Here we focus on the candidate view generation and selection pipeline, utilizing a fully trained FlexRM, fine-tuned with simulated imperfect data, as the reconstruction model. The evaluation protocol follows that outlined in section 4.2. We conduct ablation experiments by removing: (1) multi-view generation at varying elevations, (2) consistency verification (resulting in random view selection), and (3) back view generation quality assessment.

Table 4 summarizes the results, demonstrating that the removal of any of these components leads to a decrease in both CLIP (Radford et al., 2021) and VideoCLIP (Wang et al., 2023a) text similarity scores. This shows the contribution of each component in achieving high-quality 3D generation.

| Ablation | CLIP text similarity↑ | VideoCLIP text similarity↑ |
|---|---|---|
| No generation at varying elevations | 0.273 | 0.251 |
| No consistency verification | 0.269 | 0.248 |
| No back view quality assessment | 0.272 | 0.249 |
| Full model | **0.277** | **0.255** |

Table 4: **Ablation study on Candidate View Generation and Selection**. We show the results of ablating different components of our proposed candidate view generation and selection pipeline.

## 5 CONCLUSION

This paper introduces Flex3D, a novel feed-forward 3D generation pipeline that produces high-quality 3D Gaussian representations from text or single-image inputs. We propose a series of approaches to overcome the limitations of previous two-stage methods, significantly improving final 3D quality. Extensive evaluations on benchmark tasks demonstrate that Flex3D achieves state-of-the-art performance in both 3D reconstruction and generation. These results highlight the effectiveness of our approach in addressing the challenges of feed-forward 3D generation, paving the way for more robust and versatile 3D content creation.

ETHICS STATEMENT

Our work explores generative AI with a focus on generating 3D Gaussian representations from pre-existing 2D content. While we exclusively utilize ethically sourced and carefully curated training data, our model learns a generalized approach to 3D reconstruction. This means that if presented with a problematic or misleading 2D image, our model could potentially generate a corresponding 3D object, though with some reduction in quality. However, despite these inherent risks, we believe our work can empower artists and creative professionals by serving as a productivity-enhancing tool within their workflow. Furthermore, this technology has the potential to boost 3D content creation by lowering barriers to entry and providing access to individuals who lack specialized expertise.

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

## A    IMPLEMENTATION DETAILS

**Training details.** Our FlexRM is trained in a two-stage manner. In stage 1, we train it with 64 NVIDIA A100 (80GB) GPUs and use a total batch size of 512, where each batch consists of 4 multi-view images at a patch resolution of $128 \times 128$ for supervision. The input images have a resolution of $256 \times 256$, and the number of input images varies from 1 to 16. The model is trained for 10 epochs with an initial learning rate of $2 \times 10^{-4}$, following a cosine annealing schedule. Training begins with a warm-up phase of 3000 iterations, and we use the AdamW optimizer (Loshchilov & Hutter, 2017). We apply gradient clipping at 1.0 and a weight decay of 0.05, applied only to weights that are not biases or part of normalization layers. Both training and inference are performed using Bfloat16 precision. The optimization target is a combination of three different losses: L2, LPIPS, and opacity, with corresponding coefficients of 1, 2, and 1, respectively.

Stage 2 utilizes 128 NVIDIA A100 (80GB) GPUs. We increase the input image resolution to 512 $\times$ 512 and the maximum number of input images to 32. We maintain a total batch size of 512, with each batch consisting of 4 multi-view images at a resolution of $512 \times 512$ for supervision. The model is trained for 25 epochs. All other training settings including total batch size are identical to Stage 1.

For further fine-tuning using simulated imperfect input views as input, we follow the setting in Stage 2 but only train it with 32 NVIDIA A100 (80GB) GPUs for 3 epochs. We use a total batch size of 128 and an initial learning rate of $2 \times 10^{-5}$.

**3D Gaussian parameterization.** For predicted 3DGS parameters with 14 dimensions, we provide implementation details on converting them into position offset, color, opacity, scale, and rotation. We follow the setting used in GS-LRM (Zhang et al., 2024c) for opacity, scale, and rotation.

Position offset: We activate the predicted offset using a tanh function and apply a scaling factor of 0.25. This scaled offset is then added to the initial positions to obtain the final 3DGS positions.

Color: We utilize the same activation function as in Neural Radiance Fields (NeRF). The predicted color values are first passed through a sigmoid function, then multiplied by 1.002, and finally, 0.001 is subtracted. These processed values serve as zero-order Spherical Harmonics coefficients for the 3DGS.

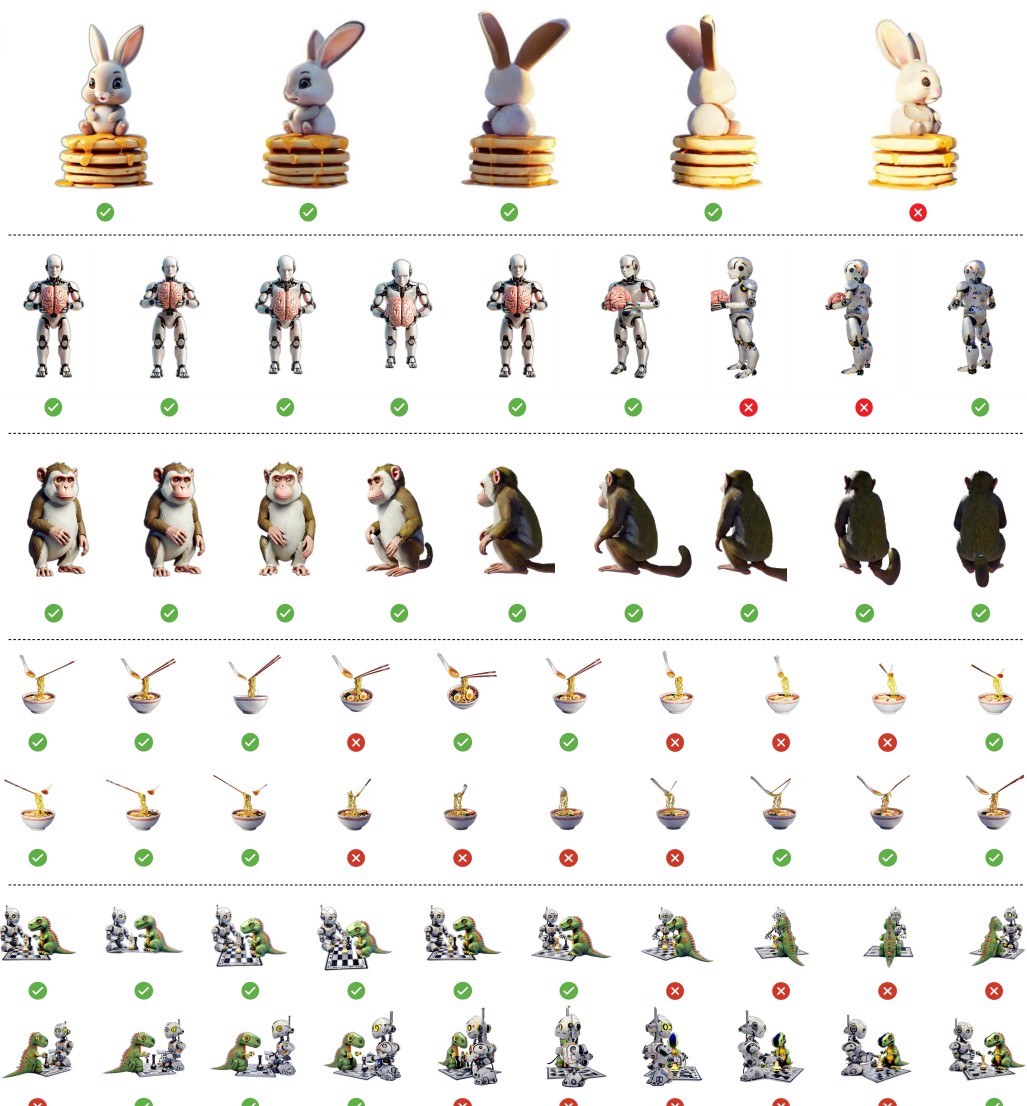

Figure 5: **View Selection Visualizations**. We show some generated candidate views for each object. A green check mark indicates that our method selected the view, while a red cross indicates that the view was rejected. As the visualization demonstrates, our method can effectively filter out views that exhibit poor quality or inconsistent results, such as those with artifacts, truncations, or awkward perspectives. This allows us to focus on reconstruction from high-quality viewpoints, leading to improved overall results.

Opacity: We subtract 2.0 from the predicted opacity before applying a sigmoid function. This approach ensures that the initial opacity values are around 0.1, which stabilizes the training process.

Scale: We subtract 2.3 from the predicted scale and then apply a sigmoid function. Additionally, we clip the scale to a maximum value of 0.3 and a minimum value of 0.0001. This design, similar to the opacity implementation, promotes stability during training.

Rotation: We predict unnormalized quaternions and apply L2-normalization as the activation function to obtain unit quaternions.

**3D Gaussian noise injection.** For all Gaussian parameters, we sample a small cube size within a range of $10 \times 10 \times 10$ to $40 \times 40 \times 40$, assuming a whole grid size of $100 \times 100 \times 100$. Each time, the size is sampled individually for every parameter to achieve greater diversity. The noise levels for

position, color, and opacity are set to 0.1, *i.e.*, a random noise between -0.1 and 0.1 is added. The noise level for scale is set to 0.02.

## B  VIEW SELECTION VISUALIZATIONS

This section provides further visualizations to illustrate the effectiveness of our view curation pipeline. Figure 5 showcases five randomly selected examples where our method successfully identifies and selects high-quality viewpoints while filtering out those with undesirable characteristics. Our method preserves high-quality views from multiple angles for objects, including front, side, and back views.

## C  ADDITIONAL EXPERIMENTS.

**Additional ablation study.** We analyze our imperfect data simulation strategy using both reconstruction and generation metrics. Evaluation settings mirror those used in the ablation study. As shown in table 5, incorporating imperfect data simulation yields improvements across both generative and reconstruction tasks. This suggests that our strategy effectively exposes the model to a wider range of data variations, enhancing its overall performance and robustness.

| Ablation | CLIP text sim↑ | VideoCLIP text sim↑ | PSNR↑ | SSIM↑ | LPIPS↓ |
|---|---|---|---|---|---|
| No simulation | 0.271 | 0.250 | 24.87 | 0.888 | 0.086 |
| Full model | **0.277** | **0.255** | **24.90** | **0.889** | **0.084** |

Table 5: **Ablation Study on Imperfect Data Simulation**. Leveraging imperfect data simulation strategy leads to a reasonable performance improvement in generative tasks and a marginal improvement in reconstruction tasks.

**Qualitative results for ablation study.** Here we present qualitative results on the effects of enabling view selection. Figure 9 demonstrates the impact of view selection on the quality of generated 3D assets. When view selection is not used, some of the generated input views in stage 1 may be less desirable. The blue circles highlight regions where deficiencies in the input result in poor generation quality. However, by incorporating view selection, the model is able to select the most high-quality and consistent views as input, generally resulting in improved 3D asset generation.

**Additional 3D reconstruction.** We additionally conduct experiments on 3D reconstruction tasks to validate FlexRM's performance across more diverse input conditions. We use 500 validation objects from our internal 3D dataset (similar to Objaverse). This validation data was held out from training. Similar to the GSO procedure, we rendered 64 views per object at four elevation degrees (-30°, 6°, 30°, and 42°), with 16 uniformly distributed azimuth degrees per elevation.

Table 6 presents the results. Similar trends are observed as with the GSO results, demonstrating the robustness of FlexRM across diverse reconstruction tasks with different data distributions.

| Method | Input views | PSNR↑ | SSIM↑ | LPIPS↓ | CLIP image sim↑ |
|---|---|---|---|---|---|
| OpenLRM | 1 | 15.41 | 0.771 | 0.241 | 0.568 |
| VFusion3D | 1 | 18.82 | 0.796 | 0.172 | 0.740 |
| FlexRM | 1 | **21.01** | **0.839** | **0.135** | **0.824** |
| InstantMesh | 4 | 20.99 | 0.822 | 0.153 | 0.782 |
| GRM | 4 | 24.65 | 0.871 | 0.124 | 0.850 |
| FlexRM | 4 | **25.32** | **0.876** | **0.083** | **0.881** |
| FlexRM | 8 | 26.11 | 0.879 | 0.080 | 0.897 |
| FlexRM | 16 | 26.29 | 0.882 | 0.079 | 0.900 |
| FlexRM | 24 | 26.42 | 0.884 | 0.078 | 0.904 |
| FlexRM | 32 | 26.54 | 0.886 | 0.077 | 0.910 |

Table 6: **Reconstruction Performance on Hand-crafted 3D Objects.** Similar to GSO reconstruction results, FlexRM still consistently outperforms other baselines across different input view settings.

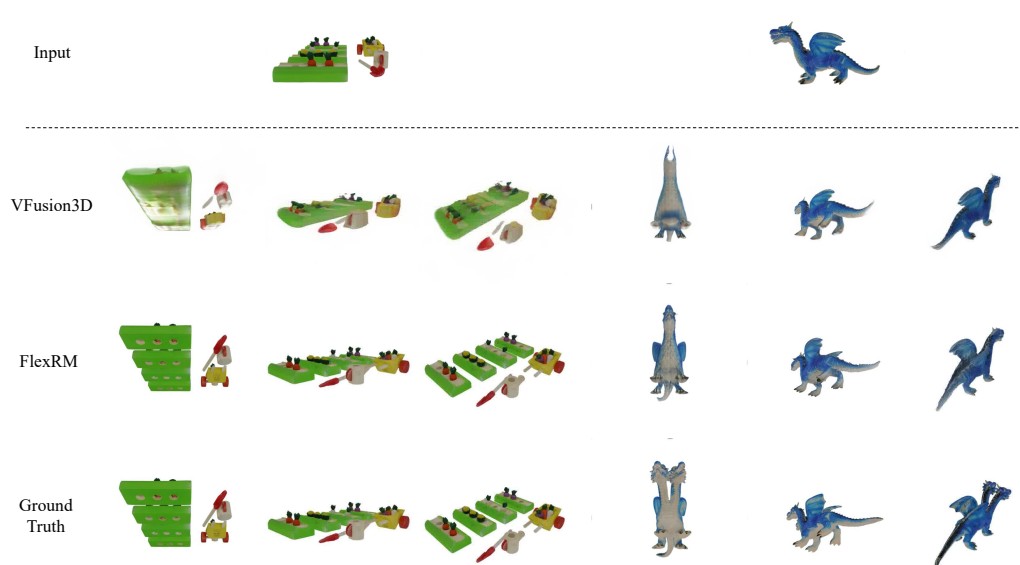

Figure 6: **Single-View Reconstruction Results**, showcasing FlexRM's ability to achieve reasonable reconstructions from only a single-view observation.

## D    QUALITATIVE RESULTS ON 3D RECONSTRUCTION

We show qualitative comparison results between FlexRM and reconstruction baselines in 1-view and 4-view settings (fig. 6 and fig. 7). FlexRM demonstrates a stronger ability to perform high-fidelity 3D reconstructions, particularly exceeding other baselines when observed from various elevation angles. This advantage is evident in both single-view and sparse-view scenarios. The results highlight FlexRM's effectiveness in capturing fine details and overall object shape, leading to more accurate and visually appealing reconstructions.

## E    LIMITATIONS

While our method can generate high-quality 3D Gaussians, the inherent problems associated with 3DGS are also present. For example, extracting clean meshes is not straightforward and usually requires multi-step post-processing. This issue can likely be mitigated in the near future given the fast development of Gaussians, either through new representations of Gaussians (Huang et al., 2024; Dai et al., 2024) or better ways to convert them to meshes (Wolf et al., 2024). Though our paper focuses on 3D object generation, another potential limitation is that the tri-plane representation is usually limited by resolution size and cannot be easily used for large scene generation.

## F    FAILURE CASES

We present some failure cases of our candidate view generation and selection pipeline in fig. 8. A notable failure occurs when the input image contains floaters or small transparent objects. This leads to incorrect generation results, especially at different elevation angles. The view selection pipeline only partially removes these incorrect results. Additionally, our view selection pipeline can sometimes produce incorrect results, particularly with objects containing thin geometries. These thin components can contribute less to feature matching, making them more susceptible to incorrect selection.

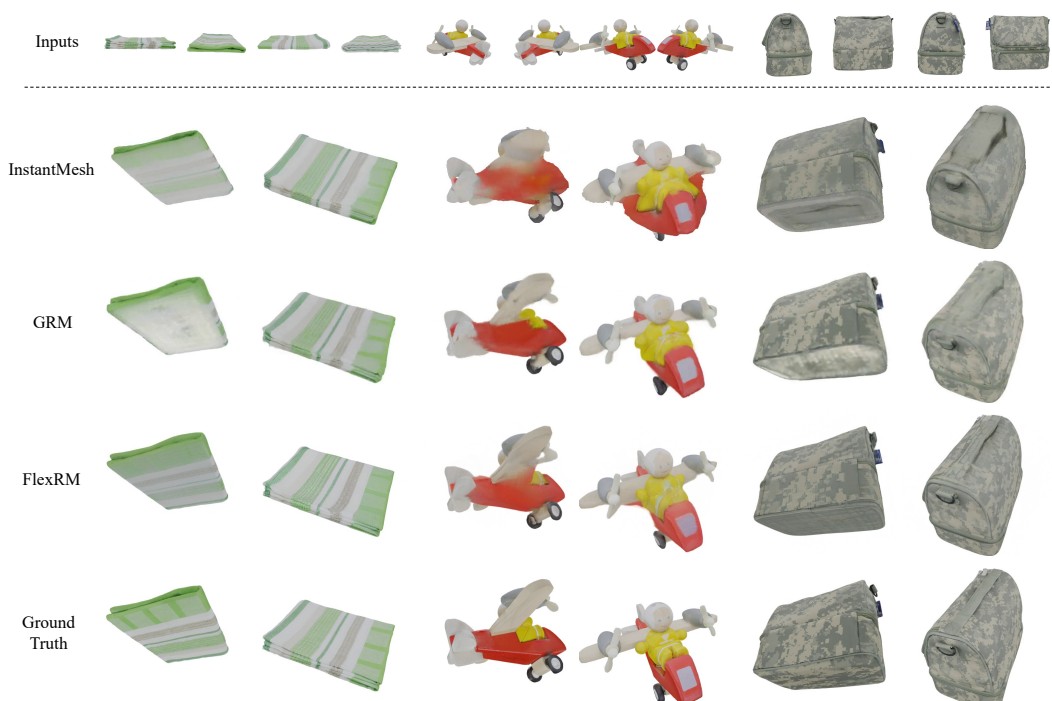

Figure 7: **4-view Reconstruction Results**. FlexRM is able to perform high-fidelity sparse-view reconstructions that closely resemble the ground truth, particularly when viewed from different elevation angles, outperforming baseline reconstructors.

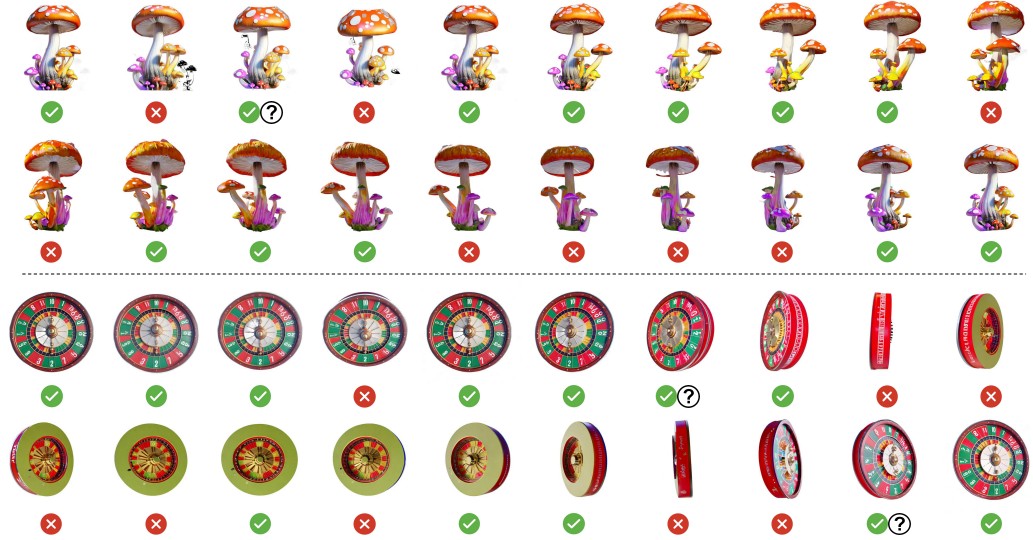

Figure 8: **Failure Cases**. A green check mark indicates that our method selected the view, while a red cross indicates that the view was rejected. Question marks indicate incorrect selection results. The top two rows show results for a mushroom, highlighting difficulties with floaters or small transparent objects inside the input image. The bottom two rows illustrates the challenges in filtering generated views of objects with thin thin geometries.

## G  IMPLICATIONS FOR FUTURE RESEARCH

**Feed-forward 3D generation.** The key insight is that we introduced a series of methods to handle imperfect multi-view synthesis results in the common two-stage 3D generation pipeline. Our whole Flex3D pipeline introduces little computational cost but yields significant performance and robustness gains, and it could serve as a common design pipeline for future research in 3D generation. Additionally, all individual components proposed in this work can be easily adopted by future research in 3D generation to improve performance. Similarly, design ideas analogous to the Flex3D pipeline could be readily adopted for large 3D scene generation.

**Feed-forward 4D generation.** Moreover, our work could be beneficial for 4D generation, which is an even more challenging task that faces similar limitations to two-stage 3D generation pipelines. Our pipeline could be directly extended to handle 4D object generation tasks. One could first generate 64 views (16 time dimensions $\times$ 4 multi-views) by fine-tuning video-based diffusion models, then slightly modify the view selection pipeline to keep only those views consistent across multiple views and time dimensions. Then, extend FlexRM from a tri-plane to a hex-plane or additionally learn time offsets to enable 4D representation. This should yield a strong method for 4D asset generation.

**Leveraging 3D understanding for generation** : Keypoint matching techniques are used in this work to effectively mitigate multi-view inconsistencies. We hope this will also inspire the 3D generation community to incorporate advanced techniques from the rapidly evolving field of 3D understanding. Recent advances in deep learning have led to significant developments in matching, tracking, deep structure from motion, and scene reconstruction. These advancements offer the 3D generation community useful tools (such as pose estimation and keypoint matching), pseudo-supervision signals (*e.g.,* pseudo-depth supervision), and new model design ideas.

## H  ADDITIONAL RELATED WORK

### H.1  MITIGATING MULTI-VIEW INCONSISTENCY WITH FEEDBACK.

Methods such as Ouroboros3D (Wen et al., 2024), Carve3D (Xie et al., 2024b), Cycle3D (Tang et al., 2024c), and IM-3D (Melas-Kyriazi et al., 2024) stem from similar motivations to our work, and they share a key idea: the feedback mechanism. While useful, these methods often require new supervision signals and learnable parameters to implement this feedback, potentially creating complex, monolithic pipelines that are difficult to decompose into reusable components for future designs. In contrast, Flex3D's components are more easily generalized. Another key difference is Flex3D's focus on the input to the reconstructor. This adds negligible computational cost and avoids the significant additional time required for multi-step refinement, preserving the speed advantage often associated with feed-forward models. Additionally, the feedback mechanism is orthogonal to our work and could be further combined with it if needed.

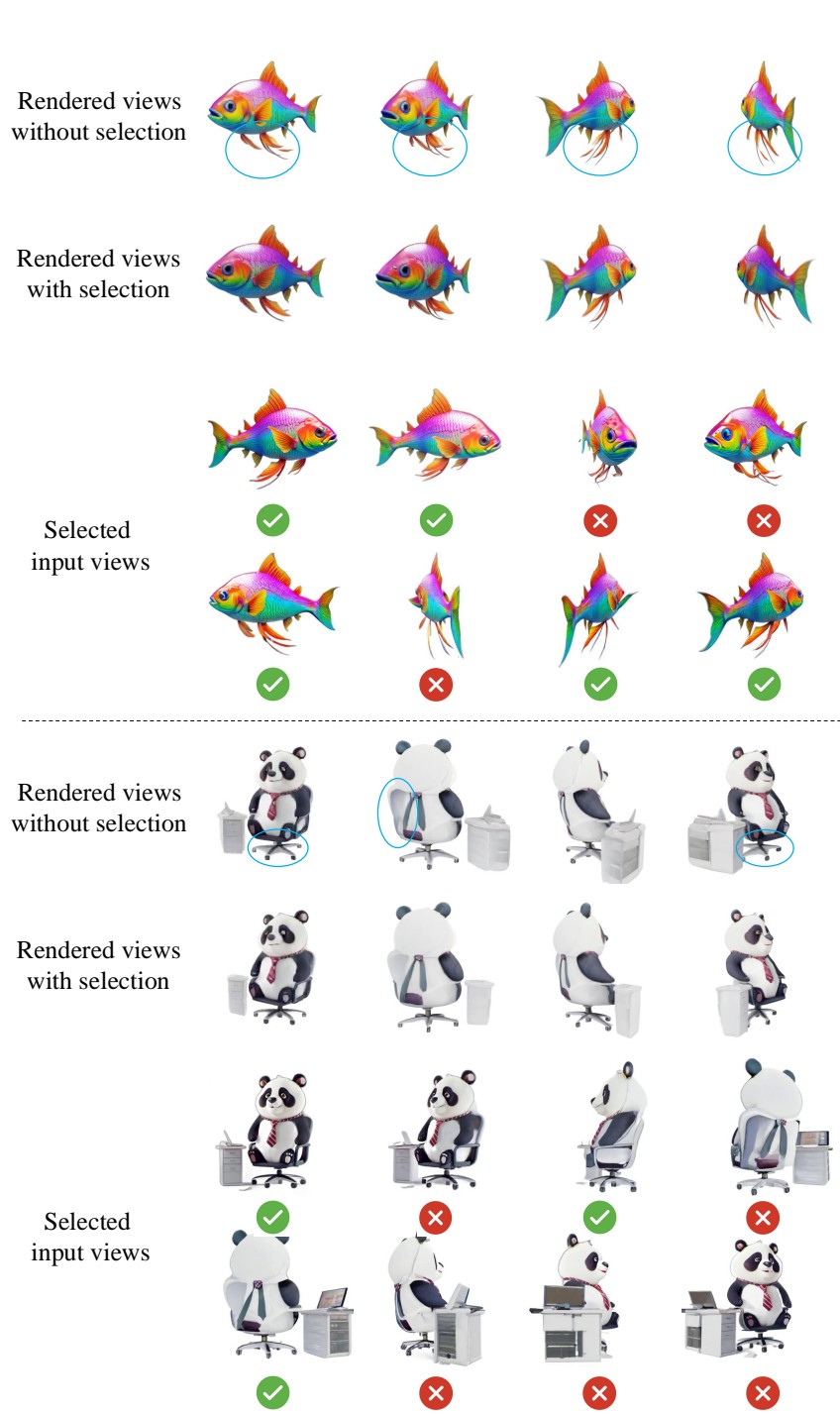

Figure 9: **Generation Results With and Without View Selection**. The top four rows show 3D generation results for a colorful rainbow fish. The first row shows the 3D assets generated by Flex3D (displayed as rendered views) without our selection pipeline. The blue circles highlight regions exhibiting generation failures. The second row presents the generated views after applying view selection. The third and fourth rows show the sampled input views (8 of 20 shown), where a green checkmark indicates that our method selected the view, and a red cross indicates that the view was rejected. The bottom four rows illustrate the same process for a different object.