# OpenReview forum: "Flex3D: Feed-Forward 3D Generation with Flexible Reconstruction Model and Input View Curation"
_ICLR.cc/2025/Conference — Submitted to ICLR 2025_

### Official Review · Reviewer_pAxN · 2024-10-30

**Soundness:** 2
**Presentation:** 2
**Contribution:** 3
**Rating:** 5
**Confidence:** 3

**Summary:**

The paper proposes Flex3D, a two-stage framework for generating high-quality 3D content using multi-view input views from a flexible reconstruction model. Initially, multiple candidate views are generated using separate multi-view diffusion models with distinct focus areas (elevation and azimuth), followed by a quality and consistency-based view selection. These selected views are then passed to a flexible reconstruction model (FlexRM), which leverages a tri-plane representation combined with 3D Gaussian Splatting (3DGS) for efficient 3D generation. Flex3D is shown to be effective in generating high-quality 3D representations and demonstrates state-of-the-art performance across several metrics in 3D generation and reconstruction tasks.

**Strengths:**

1. The two-stage process of generating and selecting views for flexible multi-view 3D generation is innovative and well-aligned with the goal of improving reconstruction quality.
2. The paper extensively validates each proposed module, demonstrating their significance through ablation studies and metrics across various tasks.

**Weaknesses:**

1. **Lack of Cohesion in Core Contributions**: The proposed approach, although effective, seems overly complex and tricky, and doesn’t clearly reflect Flex3D's core innovation. For instance, using two different models to generate two groups of multi-view images, and adding noisy inputs during reconstruction make the approach appear fragmented and difficult to generalize.
2. **Inconsistency Concerns**: The method’s use of two different models for elevation and azimuth views results in overlapping views limited to one view (that is the view with elevation of 6), raising questions about cross-model consistency. This single overlap view may not fully capture the complete object appearance, potentially leading to inconsistencies between two view sets.
3. **Inadequate Simulation of Multi-View Inconsistencies**: The noisy input augmentation during FlexRM training accounts for view quality but does not adequately model cross-view inconsistencies, due to its operation on the 3DGS.
4. **Lack of Flexibility Analysis**: The paper lacks a visual ablation study on FlexRM’s performance with varying input views to illustrate the model's robustness to input flexibility.

**Questions:**

1. Could the authors provide more insight into how the two multi-view generation models (focused on elevation and azimuth) avoid consistency issues, given the limited overlap between generated views?
2. How does FlexRM handle scenarios where significant view inconsistencies occur, especially as noisy input augmentation does not seem to address cross-view consistency?
3. Is there a visual or quantitative comparison available regarding FlexRM’s reconstruction flexibility when provided with a varying number of input views?

---

> ### Author Response · Authors · 2024-11-21
> **Thank you & responses (1)**
>
> **W1: Lack of Cohesion in Core Contributions, the approach appears fragmented and difficult to generalize.**
>
> We generally agree that Flex3D requires multiple components and can be considered complex. The Flex3D pipeline is designed to mitigate suboptimal outputs from the first-stage multi-view diffusion model. This requires considerable effort and component design to achieve high-quality text-to-3D and single-image-to-3D generation.
>
>  Although the entire pipeline might be difficult to generalize, each individual component within Flex3D can be easily generalized. For example, the minimalist design of the FlexRM architecture allows for easy implementation based on Instant-3D, and it can directly replace existing feed-forward reconstruction models.
>
> **W2 and Q1: Inconsistency Concerns regarding two different models for elevation and azimuth.**
>
> The use of two different models for elevation and azimuth leads to minimal conflict. Because there is minimal corresponding pixel overlap between the generated views, multi-view inconsistencies are unlikely, even if the two models generate different content.
>
> Furthermore, since the data used for image pre-training and fine-tuning are identical, the models are likely to generate similar content. For visual confirmation, please refer to Figure 5 (View Selection Visualizations), which presents two examples showing all 20 generated images. The results from the azimuth model are located in the top-left part (views 1-4).
>
> **W3: Inadequate Simulation of Multi-View Inconsistencies, no cross-view inconsistencies.**
>
> This is an insightful point; thank you for your careful review!  Some basic multi-view inconsistencies can be simulated by replacing some clean input images with noisy ones as final inputs. This is the strategy used in Flex3D. To simulate more substantial cross-view inconsistencies, we could inject noise multiple times, each time selecting some noisy views to include in the final input set. This approach would be fast and memory-efficient, as noise injection operates on generated Gaussian points. However, it would also add complexity to the pipeline. Therefore, we leave this as a trade-off option.
>
> **W4 and Q3: Lack of Flexibility Analysis.**
>
> Please see Table 2 for GSO reconstruction results with different numbers of input views. Overall, increasing the number of input views for FlexRM generally leads to improved reconstruction quality.  We found the improvements for object reconstruction to be less significant after 16 input views.
>
> **Q2: How does FlexRM handle scenarios where significant view inconsistencies occur?**
>
> The view selection pipeline is the primary defense against significant view inconsistencies. After this filtering stage, the input views provided to FlexRM typically contain only minor inconsistencies, which are then handled by imperfect data simulation training used in FlexRM.

---

> > ### Comment · Reviewer_pAxN · 2024-11-25
> >
> > Thank you for addressing my concerns regarding the technical details. I appreciate the detailed clarifications provided, which have helped improve my understanding of the technical aspects of your work.
> >
> > However, I still have some concerns about the **core contribution** of the paper. While I understand that the primary contribution lies in proposing the "Input View Curation" mechanism to enhance the robustness of 3D generation, I find the approach relatively straightforward. I would encourage the authors to provide a more comprehensive summary of the efforts behind their work. Additionally, as I noted in my initial comments, it would be beneficial to demonstrate the degree to which this idea directly and significantly improves the two-stage pipeline. Highlighting this impact more clearly could further strengthen the contribution.
> >
> > Moreover, I would recommend that the authors explore the potential implications of their pipeline for future research. For example, can this approach inspire advancements in more advanced pipelines such as 3D diffusion models, or in other tasks related to multi-view domain? Offering such insights could help contextualize the broader significance of this work and make me view its contributions more optimistically.
> >
> > Minor: Methods such as _Carve3D: Improving Multi-view Reconstruction Consistency for Diffusion Models with RL Finetuning_ and _Ouroboros3D: Image-to-3D Generation via 3D-aware Recursive Diffusion_ represent two categories of approaches that align with your motivation. I suggest that the authors consider including a discussion on these methods in the paper. A theoretical comparative analysis highlighting the strengths and weaknesses of your approach relative to these methods could provide valuable insights into the two-stage pipeline's potential in 3D generation. Given time constraints, the authors may choose not to revise the paper immediately but could address this in future iterations.
> >
> > Thank you again for your efforts in improving the paper. I look forward to seeing how these aspects are addressed.

---

> > > ### Author Response · Authors · 2024-11-26
> > > **Thank you & responses (2)**
> > >
> > > Thank you for your further thorough and insightful review of our paper and rebuttal! We appreciate you taking the time to provide such detailed feedback, and we especially value your constructive criticism regarding the core contribution and its broader implications. We are very glad that concerns regarding the technical details are well resolved. Here we provide our replies to your kind suggestions:
> > >
> > > **S1: I would encourage the authors to provide a more comprehensive summary of the efforts behind their work.**
> > >
> > > We first summarize our contributions: Two-stage 3D generation pipelines, such as Instant3D and many others, are the most popular frameworks for text- or single-image-based 3D generation. However, a significant limitation of all current approaches is that while their reconstructors perform well with sparse-view reconstruction, the final 3D quality remains constrained by the quality of the generated multi-view images. We propose a series of approaches, including (1) candidate view generation and curation, (2) a flexible view reconstruction model, and (3) noise simulation, to gradually address this challenge. Specifically, (1) directly addresses the challenge of mitigating suboptimal outputs from the first-stage multi-view diffusion model; (2) enables high-quality feed-forward reconstruction from selected views; and (3) enhances the reconstruction model's robustness against small noise, further improving the final 3D quality.
> > >
> > > We have also revised our manuscript in the end of introduction (summary of contributions) and conclusion sections to further clarify our contributions.
> > >
> > > **S2: Highlighting this impact (view selection) more clearly could further strengthen the contribution.**
> > >
> > > Although we have presented detailed ablation studies demonstrating the effectiveness of the view selection pipeline, only qualitative results were reported, and their impact may not have been sufficiently emphasized. To more directly highlight the contribution of the proposed view selection pipeline, we have included additional qualitative results in the Appendix (Section C
> > > and Figure 9) of the revised manuscript.
> > >
> > > The conclusion is: When view selection is not used, some of the generated input views in stage 1 may be less desirable, leading to poorer quality 3D asset generation. However, by incorporating view selection, the model can select the most high-quality and consistent views as input, generally resulting in improved 3D asset generation.

---

> > > > ### Author Response · Authors · 2024-11-26
> > > > **Thank you & responses (3)**
> > > >
> > > > **S3: I would recommend that the authors explore the potential implications of their pipeline for future research.**
> > > >
> > > > We agree that highlighting the potential impact of our work on future research would be highly beneficial! We expand the discussion here to include the following topics:
> > > >
> > > > **Feed-forward 3D generation:** We anticipate that two-stage 3D generation pipelines will remain popular in the future due to their many advantages. For example, they can easily adopt pre-trained diffusion models, and sparse-view inputs greatly simplify the reconstruction process, often leading to the best results. This line of research can draw many useful implications from our work, which makes the question we are addressing even more important.
> > > >
> > > > The key insight is that we introduced a series of methods to handle imperfect multi-view synthesis results in the common two-stage 3D generation pipeline. Our whole Flex3D pipeline introduces little computational cost but yields significant performance and robustness gains, and it could serve as a common design pipeline for future research in 3D generation. Additionally, all individual components proposed in this work can be easily adopted by future research in 3D generation to improve performance.   Similarly, design ideas analogous to the Flex3D pipeline could be readily adopted for large 3D scene generation.
> > > >
> > > > **Feed-forward 4D generation:** Moreover, our work could be beneficial for 4D generation, which is an even more challenging task that faces similar limitations to two-stage 3D generation pipelines. Our pipeline could be directly extended to handle 4D object generation tasks. One could first generate 64 views (16 time dimensions * 4 multi-views) by fine-tuning video-based diffusion models, then slightly modify the view selection pipeline to keep only those views consistent across multiple views and time dimensions. Then, extend FlexRM from a tri-plane to a hex-plane or additionally learn time offsets to enable 4D representation. This should yield a strong method for 4D asset generation.
> > > >
> > > > **Leveraging 3D understanding for generation:** Keypoint matching techniques are used in this work to effectively mitigate multi-view inconsistencies. We hope this will also inspire the 3D generation community to incorporate advanced techniques from the rapidly evolving field of 3D understanding. Recent advances in deep learning have led to significant developments in matching, tracking, deep structure from motion, and scene reconstruction. These advancements offer the 3D generation community useful tools (such as pose estimation), pseudo-supervision signals (e.g., pseudo-depth supervision), and new model design ideas.
> > > >
> > > > We have also included discussions here in the appendix of the revised manuscript (section G).
> > > >
> > > > **S4: I suggest that the authors consider including a discussion on these methods (Carve3D and Ouroboros3D) in the paper.**
> > > >
> > > > Thank you for your suggestion! These points are certainly worth discussing, as they stem from similar motivations to our work. We have added a related work section to the revised manuscript (Section H), which is currently placed in the appendix to avoid major changes to the main paper. This allows reviewers to easily locate the referenced tables and figures. We will move this section to the main body of the paper in the camera-ready version.
> > > >
> > > > Our discussion is as follows: Although Ouroboros3D emphasizes 3D asset reconstruction and Carve3D focuses on multi-view generation, these methods, along with others like Cycle3D [1] and IM-3D [2], share a key idea: the feedback mechanism. While useful, these methods often require new supervision signals and learnable parameters to implement this feedback, potentially creating complex, monolithic pipelines that are difficult to decompose into reusable components for future designs. In contrast, Flex3D's components are more easily generalized. Another key difference is Flex3D's focus on the input to the reconstructor. This adds negligible computational cost and avoids the significant additional time required for multi-step refinement, preserving the speed advantage often associated with feed-forward models. Additionally, the feedback mechanism is orthogonal to our work and could be further combined with it if needed.
> > > >
> > > > [1] Cycle3D: High-quality and Consistent Image-to-3D Generation via Generation-Reconstruction Cycle, arXiv 2024.
> > > >
> > > > [2]: IM-3D: Iterative Multiview Diffusion and Reconstruction for High-Quality 3D Generation, ICML 2024.

---

> > > > > ### Comment · Reviewer_pAxN · 2024-12-03
> > > > >
> > > > > Thank you for your rebuttal. I appreciate the insights and contributions your work brings to other areas, which I find meaningful. However, I remain concerned about the necessity of the "Input View Curation" mechanism in your approach for future research, as well as the significance of the performance improvements it offers. Consequently, I consider this work to be on the borderline, with both borderline acceptance and borderline rejection being possible outcomes. Therefore, I have decided to maintain my original score while reducing my confidence in the evaluation.

---

> ### Author Response · Authors · 2024-12-03
>
> Thank you for your thorough review and for taking the time to consider our rebuttal, especially for a second time! We appreciate your acknowledgment of the potential contributions our work offers to other areas, and we're glad you find that aspect meaningful.
>
> We understand your remaining concerns regarding the necessity and impact of the "Input View Curation" mechanism. We are offering this further response **not to** change your score, but rather to further engate in the discussion and perhaps offer insights for future work in this research area. Please feel free to discuss any topics in this research area further.
>
> We agree that if multi-view diffusion models could synthesize **perfectly 3D-consistent views**, the view selection pipeline would become unnecessary. However, we believe that until such models are widely available and robust—and considering the limitations we've observed in current multi-view diffusion methods (even across various 4-view diffusion models)—our view curation pipeline remains valuable. We envision it as a practical and beneficial tool for at least the next 3 years, particularly because it can also serve 4D generation frameworks that require consistency in both 3D and temporal dimensions.
>
> Regarding performance gains, while intuitively the improvement from view selection might diminish with future, stronger multi-view diffusion models (although the threshold could be adapted to retain more high-quality views), we have found it to be meaningful across different reconstruction models and video diffusion models. This broader applicability suggests a benefit **beyond the specific models developed in this paper**. It could also be very beneficial for generative reconstruction frameworks such as Im-3D, Cat3D, and Cat4D, where inconsistent views should be filtered out before fitting a 3D/4D representation. We would like to explore this further in future work.
>
> Thank you again for your time and insightful feedback. We greatly appreciate your review and engagement, and we truly enjoyed discussing this work and border the 3D generation research area with you!

---

### Official Review · Reviewer_9shi · 2024-11-01

**Soundness:** 3
**Presentation:** 3
**Contribution:** 3
**Rating:** 6
**Confidence:** 5

**Summary:**

The author follow the classical two-stage 3D generation model: 1) Multi-view Generation; 2) A Large Gaussian reconstruction model conditioned on multi-view images from stage one to generate 3D Gaussian model. The author present a simple but effective sample strategy to choose some high-quality multi-view images among generated images as the inputs for the reconstruction stage.

**Strengths:**

1) The paper is well-written and easy to follow.

2) The results the author shows are compelling.

3) The chosen multi-view strategy for improved quality is somewhat new.

**Weaknesses:**

The multi-view generation model is too heavy. It requires two multi-view generations to produce dense-view proposals. I believe this process is time-consuming and memory-intensive. What is the inference time required to produce the multi-view image proposals? Does it possible to apply video diffusion mode to generate  a trajectory where elevation varies according to the sine function, and azimuth is selected at equal intervals instead of two multi-view diffusion model?

**Questions:**

1) Please show me some failure cases, especially for your view-selection method that failed.


2) Missing some Reference:

[1] Li W, Chen R, Chen X, et al. Sweetdreamer: Aligning geometric priors in 2d diffusion for consistent text-to-3d[J]. arXiv preprint arXiv:2310.02596, 2023.
[2] Qiu L, Chen G, Gu X, et al. Richdreamer: A generalizable normal-depth diffusion model for detail richness in text-to-3d[C]//Proceedings of the IEEE/CVF Conference on Computer Vision and Pattern Recognition. 2024: 9914-9925.
[3] Chen R, Chen Y, Jiao N, et al. Fantasia3d: Disentangling geometry and appearance for high-quality text-to-3d content creation[C]//Proceedings of the IEEE/CVF international conference on computer vision. 2023: 22246-22256.

---

> ### Author Response · Authors · 2024-11-21
> **Thank you & responses (1)**
>
> **W: What is the inference time required to produce the multi-view image proposals? Does it possible to apply video diffusion mode to generate a trajectory where elevation varies according to the sine function, and azimuth is selected at equal intervals instead of two multi-view diffusion model?**
>
> While view selection and FlexRM reconstruction require less than a second on a single A100 GPU, generating 20 views with two diffusion models takes approximately one minute on a single H100 GPU. This speed is similar to video-based multi-view diffusion models like SV3D. We have added a note regarding this in the revised paper.
>
> We agree with reviewer 9shi that applying video diffusion models to generate a trajectory where elevation varies according to the sine function, and azimuth is selected at equal intervals, is both sound and feasible. Thank you for this suggestion! We intend to leave this promising idea in future work.
>
> **Q1: Please show me some failure cases, especially for your view-selection method that failed.**
>
> We have added failure cases in Figure 8 and Section F in the appendix (page 20) of the revised manuscript. A notable failure occurs when the input image contains floaters or small transparent objects. This leads to incorrect generation results, especially at different elevation angles. The view selection pipeline only partially removes these incorrect results. Additionally, our view selection pipeline can sometimes produce incorrect results, particularly with objects containing thin geometries. These thin components can contribute less to feature matching, making them more susceptible to incorrect selection.
>
> **Q2: Missing some Reference.**
>
> Thank you for your suggestions! We have added all of them in the revised manuscript.

---

> > ### Author Response · Authors · 2024-11-28
> > **Gentle reminder**
> >
> > Dear Reviewer 9shi:
> >
> > We sincerely appreciate the time and effort you dedicated to reviewing our paper! In response to your concerns, we have reported failure cases with detailed analysis during the discussion period.
> >
> > As the discussion period concludes soon, we kindly request that, if possible, you review our rebuttal at your convenience. Should there be any further points requiring clarification or improvement, we are fully committed to addressing them promptly. Thank you once again for your invaluable contribution to our manuscript!
> >
> > Warm regards,
> > The Authors

---

> > ### Comment · Reviewer_9shi · 2024-12-03
> >
> > Thanks for the response for the original questions. The rebuttal well addressed my comments and questions. I keep my original score.

---

> ### Author Response · Authors · 2024-12-03
>
> Thank you for your time and consideration in reviewing our manuscript and rebuttal. We appreciate your feedback and are very pleased to hear that the rebuttal adequately addressed your comments and questions!

---

### Official Review · Reviewer_f8fj · 2024-11-03

**Soundness:** 3
**Presentation:** 3
**Contribution:** 3
**Rating:** 6
**Confidence:** 3

**Summary:**

The paper proposes a robust feedforward 3D generation pipeline to address inconsistent multiview inputs. Specifically, it fine-tunes multiview and video diffusion models to generate diverse viewing angles and incorporates a key view selection module using an existing feature-matching model. This approach ensures that high-quality and consistent views are chosen for 3D reconstruction.

**Strengths:**

1. Logical Model Design and Well-organized Writing. Good logical model design and clarity of writing. It effectively identifies the limitations of existing models and systematically addresses them step by step, making the problem-solving process easy for readers to follow. This demonstrates a well-structured research design, facilitating readers’ comprehension of the methodology and approach.

2. Practicality. The techniques for multi-view generation, view selection, and robustness through data augmentation provide substantial applicability and reusability. The paper builds on an existing Instant3D architecture and employs a systemically optimized approach, suggesting high utility. It would be beneficial If authors release all the pre-trained models.

**Weaknesses:**

1. Incremental technical improvement. The suggested pipeline combines and optimizes existing techniques rather than introducing innovative algorithms. The approach appears to rely heavily on integrating and optimizing pre-existing technologies rather than presenting a novel concept or unique contribution.

2. Complex Pipeline Requiring Extensive Fine-Tuning and Training. While the pipeline is logically structured, it is complex and demands extensive fine-tuning and training. Five rounds of fine-tuning are required. Initial multi-view generation involves data creation and two rounds of fine-tuning. The view selection step also utilizes existing models to build a new system module. Subsequently, the feed-forward model undergoes two additional rounds of fine-tuning, and the process includes one more phase of training with data augmentation. This level of complexity could hinder full implementation and reproducibility.

3. Performance Concerns Relative to Complexity. Given the overall complexity, the proposed model’s 3D generation performance shows rather minor improvements. For instance, as shown in Table 1, the performance metrics are slightly higher than those of other models.

**Questions:**

1. If the model is designed to be robust across various poses, view counts, and noise levels, could you provide visual results demonstrating this? For example, does the model perform well when given a side or back view as a single input? Additionally, how much inconsistency can be tolerated during the multi-view selection process?

2. Does the performance continue to improve as the number of views increases? How does the processing time scale with more views? If more views are beneficial, what strategies could be used to efficiently handle a greater number of input views?

3. It could be confusing if the notation for f in line 294 differs from f in line 288.

4. Where are the results for the 32-view test reported in line 489?

5. What would the outcome be if the selected views were used for a NeRF-based approach, similar to Instant3D? While GS may be preferred for faster training, NeRF could potentially yield better final performance.

6. Why are the two-stage training and imperfect input view simulation conducted as separate processes?

---

> ### Author Response · Authors · 2024-11-21
> **Thank you & responses (1)**
>
> **W1: Lack of novel concept or unique contribution.**
>
> We acknowledge similarities with previous works, especially given the rapid advancements in 3D feed-forward models since LRM, which, despite being just published at ICLR 2024, has already garnered over 214 citations according to Semantic Scholar.
> We'd like to address the comment regarding  the lack of novel concept or unique contribution. The starting point of our work stems from a key limitation not yet addressed in existing research. Two-stage 3D generation pipelines, like Instant3D and many others [1-10], are currently the most popular framework. However, a significant limitation of all these approaches is that while their reconstructors perform well with sparse-view reconstruction, the final 3D quality remains constrained by the quality of the generated multi-views. Our work directly addresses the challenge of mitigating suboptimal outputs from the first-stage multi-view diffusion model. We incorporate view selection, a flexible view reconstruction model, and noise simulation to resolve this issue, crucial for text-to-3D and single-image-to-3D generation. This is the core idea we want to convey, and we believe it is a novel concept.
>
> [1] Instant3d: Fast text-to-3d with sparse-view generation and large reconstruction model. ICLR 2024.
>
> [2] GRM: Large Gaussian Reconstruction Model for Efficient 3D Reconstruction and Generation. ECCV 2024.
>
> [3] GS-LRM: Large Reconstruction Model for 3D Gaussian Splatting. ECCV 2024.
>
> [4] LGM: Large Multi-view Gaussian Model for High-Resolution 3D Content Creation. ECCV 2024.
>
> [5] CRM: Single Image to 3D Textured Mesh with Convolutional Reconstruction. ECCV 2024.
>
> [6] Meta 3D AssetGen: Text-to-Mesh Generation with High-Quality Geometry, Texture, and PBR Materials. NeurIPS 2024.
>
> [7] LRM-Zero: Training Large Reconstruction Models with Synthesized Data. NeurIPS 2024.
>
> [8] GeoLRM: Geometry-Aware Large Reconstruction Model for High-Quality 3D Gaussian Generation. NeurIPS 2024.
>
> [9] InstantMesh: Efficient 3D Mesh Generation from a Single Image with Sparse-View Large Reconstruction Models. arXiv 2024.
>
> [10] Tencent Hunyuan3D-1.0: A Unified Framework for Text-to-3D and Image-to-3D Generation. arXiv 2024.
>
> **W1: The approach appears to rely heavily on integrating and optimizing pre-existing technologies.**
>
> Regarding specific technical components, we highlight our key differences and improvements compared to previous work:
>
> **FlexRM (Stage 2)**: Previous works [11,12] rely on a position prediction branch to determine 3D Gaussian positions, but FlexRM can directly determine 3D Gaussian positions from the tri-plane representation using our proposed designs. Furthermore, FlexRM can process up to 32 input views and demonstrates significantly stronger performance than [11,12].
>
> **Multi-view image generation (Stage 1)**: Our contribution involves generating novel views with separate models for elevation and azimuth angles, minimizing multi-view conflicts and achieving better results compared to single-model approaches like SV3D [13] (We are happy to provide further visual results if necessary). Besides this, our proposed view curation pipeline effectively removes suboptimal or 3D-inconsistent generated views, improving the final 3D asset quality.
>
> **Camera Conditioning (Stage 2)**: This minor design choice improves handling of multiple input views, especially since reconstruction models are trained with a varying number of them. As shown in lines 500-504 in the revised manuscript, the benefits of stronger camera conditioning become more effective with a larger number of input views. This modification is very simple and introduces negligible computational overhead.
>
> **Imperfect Data Simulation (Stage 2)**: This novel contribution enhances FlexRM's robustness to minor imperfections in generated multi-view images, improving performance, particularly in generative tasks. While gains are marginal for reconstruction tasks (Table 5, right side), they are non-marginal for generation tasks (Table 5, left side), where the CLIP text similarity score increases from 27.1% to 27.7%. This metric is typically very close (<2% difference) among different methods.
>
> [11] Triplane Meets Gaussian Splatting: Fast and Generalizable Single-View 3D Reconstruction with Transformers, CVPR 2024.
>
> [12] AGG: Amortized Generative 3D Gaussians for Single Image to 3D, TMLR 2024
>
> [13] SV3D: Novel Multi-view Synthesis and 3D Generation from a Single Image using Latent Video Diffusion, ECCV 2024.
>
> **W2: Complex Pipeline Requiring Extensive Fine-Tuning and Training.**
>
> We fairly agree the pipeline does require extensive fine-tuning. Other than fine-tuning, the FlexRM architecture is designed with a minimalist philosophy and can be easily reproduced based on Instant-3D. We have included details of FlexRM training at all stages to facilitate full reimplementation. For multi-view generation, SV3D can be used as an alternative, though performance may be slightly reduced.

---

> ### Author Response · Authors · 2024-11-21
> **Thank you & responses (2)**
>
> **W3: Performance Concerns Relative to Complexity.**
>
> Since CLIP-based metrics can be unstable for 3D evaluation, we place greater emphasis on the user study results and qualitative comparisons. Our method outperformed all baselines with at least a 92.5% win rate in the user study. Furthermore, our anonymous webpage showcases over 60 3D videos and 15 interactive 3D Gaussian Splatting results, demonstrating the qualitative strengths of our approach. We are happy to provide further comparison videos with baselines in the anonymous webpage if needed.
>
> **Q1: Results on robustness across various poses, view counts, and noise levels.**
>
> **Poses:** To test this, we ran a GSO single-view reconstruction experiment with different input viewing angles. In addition to the front view reported in the main paper, we tested left, right, and back views. The results below show that FlexRM is robust to different poses and achieves consistently good performance.
>
> | Pose | PSNR&uarr; | SSIM&uarr; | LPIPS&darr; | CLIP image sim&uarr;|
> |---|---|---|---|---|
> | Front | 21.21 | 0.862 | 0.125 | 0.832 |
> | Left | 21.25 | 0.862 | 0.126 | 0.831 |
> | Right | 21.18 | 0.863 | 0.127 | 0.831 |
> | Back | 21.38 | 0.863 | 0.126 | 0.832 |
>
>
> **View counts:**  For more results on varying view counts (single-view, 4-view, 8-view, 16-view, 24-view, and 32-view), please refer to Table 2. Overall, FlexRM exhibits robustness with different numbers of input views, and increasing the number of input views generally leads to improved reconstruction quality.
>
>
> **Noise levels:** Robustness to noise was evaluated using GSO 4-view reconstruction. Different levels of Gaussian noise (standard deviations of 0, 0.01, 0.05, and 0.1) were added randomly to two of the input images. The results demonstrate that FlexRM is robust to small amounts of noise and maintains good performance even with moderate noise levels:
>
> | Noise level (σ) | PSNR&uarr; | SSIM&uarr; | LPIPS&darr; | CLIP image sim&uarr; |
> |---|---|---|---|---|
> | 0 | 25.55 | 0.894 | 0.074 | 0.893 |
> | 0.01 | 25.53 | 0.893 | 0.074 | 0.892 |
> | 0.05 | 25.17 | 0.884 | 0.078 | 0.884 |
> | 0.1 | 24.37 | 0.877 | 0.084 | 0.873 |
>
> The presented GSO reconstruction results should be sufficient to demonstrate the robustness of FlexRM. Qualitative results can be provided in the supplementary material if needed.
>
> **Q1:  How much inconsistency can be tolerated during the multi-view selection process?**
>
> We set a hyperparameter (a matching point count threshold) to make our view selection pipeline compatible with different multi-view generative models. The default threshold is the mean minus 0.6 times the standard deviation (as described in lines 250-252 of the main paper). This threshold generally balances removing significantly inconsistent views and retaining those with minor imperfections. It can be adjusted. A lower threshold is more lenient, keeping more views (and thus tolerating more inconsistency). A higher threshold is stricter, discarding more views to ensure greater consistency among the selected subset.
>
> **Q2 and Q4: Does the performance continue to improve as the number of views increases? How does the processing time scale with more views? What strategies could be used to efficiently handle a greater number of input views? Where are the results for the 32-view test reported in line 489?**
>
> We have added 24-view and 32-view results to Table 2 in the revised manuscript. We found the improvements for object reconstruction to be less significant after 16 input views. Processing time scales only slightly with the number of input views; even with 32 views, FlexRM can still generate 1M 3D Gaussian points in less than a second. To further enhance efficiency, one strategy could be to redesign the network architecture, for instance, by using SSMs (State-space models) to replace the transformer network.
>
> **Q3: It could be confusing if the notation for f in line 294 differs from f in line 288.**
>
> Thank you for pointing this out! We have added the missing MLP to the equation in the revised manuscript to avoid ambiguity.
>
> **Q5:  What would the outcome be if the selected views were used for a NeRF-based approach, similar to Instant3D?**
>
> Using a tri-plane NeRF, compared to our designed direct tri-plane 3DGS, results in performance degradation. In GSO single-view reconstruction performance degrades by approximately 2 dB PSNR. For 4-view sparse reconstruction, the degradation is more substantial, approximately 3 dB PSNR. This highlights the effectiveness of our direct tri-plane 3DGS design.
>
> **Q6: Why are the two-stage training and imperfect input view simulation conducted as separate processes?**
>
> This is an interesting point! Two-stage training and imperfect input view simulation can indeed be combined. Separating them allows the simulation to be optional. This is beneficial for models focused solely on reconstruction, where simulating imperfect views isn't necessary.

---

> > ### Comment · Reviewer_f8fj · 2024-11-25
> >
> > Thank you, authors, for your detailed and thoughtful responses. After revisiting the paper and carefully considering your clarifications, as well as feedback from other reviewers, I have decided to maintain my score of 6.
> >
> > The approach is systematically designed and demonstrates meaningful improvements. Based on your responses, the proposed FlexRM pipeline more clearly addresses limitations in multi-view diffusion models and reconstruction methods by integrating view selection, a tri-plane-based reconstruction model, and imperfect data simulation to enhance robustness and consistency. However, the contributions primarily focus on optimizing existing methods rather than introducing fundamentally novel concepts. While your clarifications provide valuable insights, questions remain regarding the scalability and broader applicability of the complex multi-stage pipeline, which requires extensive fine-tuning. These considerations lead me to retain my initial assessment.

---

> ### Author Response · Authors · 2024-11-25
>
> We sincerely appreciate the time and effort you invested in providing such detailed and thoughtful feedback on our manuscript! We are grateful for your careful reconsideration of our paper, taking into account both our rebuttal and the other reviews. We are pleased that most concerns have been well-addressed by the rebuttal and that valuable insights have also been provided.
>
> We understand and concur with your assessment regarding the complexity of the pipeline. We will keep these points in mind as we move forward with future iterations of this work. Thank you once again for your invaluable feedback!

---

> ### Author Response · Authors · 2024-11-26
> **Re-consider confidence score**
>
> Dear Reviewer f8fj,
>
> We apologize for the inconvenience. We would like to thank you once again for carefully reviewing our submission, rebuttal, and the other reviews! Your review clearly reflects a very good understanding of this submission and related works, and we find many of your questions insightful. We kindly ask if you would consider reconsidering your confidence score. We feel a confidence score of 3 does not fully reflect your expertise in this area and the effort you put into reviewing this submission.
>
> Warm regards,
> The Authors

---

### Official Review · Reviewer_oAc3 · 2024-11-04

**Soundness:** 3
**Presentation:** 2
**Contribution:** 1
**Rating:** 5
**Confidence:** 4

**Summary:**

This paper proposes Flex3D, a method for feed-forward 3D generation. The method is split into two stages, i.e., multi-view generation and subsequent conversion of these generated multi-view images into 3D Gaussians for arbitrary view rendering. The first stage uses a multi-view image model and a image-to-video model to generate multiple viewpoints of a scene. The second stage uses a LRM-like pipeline to generate 3D Gaussians. The results show competitive quality compared to previous works.

UPDATE: given the additional experiments provided in the rebuttal that show how the proposed tweaks can improve other models, I go back from "reject" to my original rating of "marginally below". This partially resolves the closed-source issue (in response to [this](https://openreview.net/forum?id=2vaTZH31oR&noteId=nksfmi6e00). Nonetheless, the extent of contributions is still small, and this remains a borderline paper.

**Strengths:**

- Visual quality: the results look good and similar/slightly better than previous works
- Back view quality assessment: using a multi-view video classifier to tackle typically lower back-facing views generation seems interesting, even though little information is provided.

**Weaknesses:**

- There is a general lack of technical insights
- FlexRM stage already proposed (Stage 2): Previous works [1,2] in feed-forward 3D generation already proposed last year to decode triplane features into 3D Gaussian attributes.
- Multi-view image generation already proposed (Stage 1): MVDream [3] and follow-up works already turn pre-trained image generators into multi-view generators.
- Multi-view image generation with video model already proposed (Stage 1): Previous works [4,5] already proposed to use video generators for novel view synthesis given an image as an input.
- Conditioning with camera already proposed and marginal (Stage 2): previous works such as SV3D [5] already proposed to condition the generation with camera matrices. In this work it is used in the image encoder DINO. However, the ablation in Tab. 3 shows that the model with “No stronger camera cond” only shows very marginal improvement?
- Imperfect data simulation with marginal improvements (Stage 2): the data simulation part in the method section sounds rather complicated and unnecessary given its little impact in Tab. 5? Similar to the camera conditioning, the metrics only show very marginal improvement?
- No computational cost analysis: The method seems very complicated, it would be good to compare training and inference time against previous works.

References:
- [1] Zou et al., Triplane Meets Gaussian Splatting: Fast and Generalizable Single-View 3D Reconstruction with Transformers, arXiv 2023
- [2] Xu et al., AGG: Amortized Generative 3D Gaussians for Single Image to 3D, TMLR 2024
- [3] Shi et al., MVDream: Multi-view Diffusion for 3D Generation, ICLR 2024
- [4] Kwak et al., ViVid-1-to-3: Novel View Synthesis with Video Diffusion Models, CVPR 2024
- [5] Voleti et al., SV3D: Novel Multi-view Synthesis and 3D Generation from a Single Image using Latent Video Diffusion, ECCV 2024

**Questions:**

I don’t really have technical questions, and it is rather unlikely that I will change my score (I hesitated between 5:weak reject and 3:reject).
This is because, while the quality of writing is decent and results marginally improve on the state of the art, the paper reads more like a white-paper for re-engineering a large-scale system rather than answering any specific scientific question.

What are the **insights** that were not proposed before that could be adopted in follow-up research?
Or is this work just about combining previous techniques with the sole purpose of getting (very marginal) improvements to metrics?

And given the metrics improvements are so marginal (as revealed by the ablations), why does all of this complication really matter?
Perhaps the small improvement in metrics does not reflect a drastic improvement in qualitative performance… but I wasn’t able to see a drastic improvement in qualitative results on the supplementary website… so I am having a very hard time to consider all the proposed complications to be truly worth it.

For a system paper that needs 128 A100 to train, I would have expected a **much** larger improvement in performance to justify a white-paper as a technical conference paper. The story would be different if the pre-trained model and/or code+data was released, and the method tested on public benchmarks.

**Details Of Ethics Concerns:**

You mention all data was "ethically sourced"... but a pointer to a study that confirms that this is the case would be good to add. But how can the reader be confident this is the case... given the dataset is internal and will not be released? And what does ethically sourced really mean...?
Did you pay the 3D artists individually for the models used, or did you just scrape data from web repos?

---

> ### Author Response · Authors · 2024-11-21
> **Thank you & responses (1)**
>
> **W1: There is a general lack of technical insights.**
>
> We acknowledge similarities with previous works. This is hard to avoid, especially given the rapid progress in 3D feed-forward models since LRM (ICLR 2024), which already has over 214 citations according to Semantic Scholar.
>
> We'd like to address the comment regarding a lack of insights. The starting point of our work stems from a key limitation not yet addressed in existing research. Two-stage 3D generation pipelines, like Instant3D and many others [1-10], are currently the most popular framework and generally achieve the best performance. However, a significant limitation of all these approaches is that while their reconstructors perform well with sparse-view reconstruction, the final 3D quality remains constrained by the quality of the generated multi-views. Our work directly addresses the challenge of mitigating suboptimal outputs from the first-stage multi-view diffusion model. We incorporate view selection, a flexible view reconstruction model, and noise simulation to resolve this issue, crucial for text-to-3D and single-image-to-3D generation. This is the core insight we want to deliver.
>
> [1] Instant3d: Fast text-to-3d with sparse-view generation and large reconstruction model. ICLR 2024.
>
> [2] GRM: Large Gaussian Reconstruction Model for Efficient 3D Reconstruction and Generation. ECCV 2024.
>
> [3] GS-LRM: Large Reconstruction Model for 3D Gaussian Splatting. ECCV 2024.
>
> [4] LGM: Large Multi-view Gaussian Model for High-Resolution 3D Content Creation. ECCV 2024.
>
> [5] CRM: Single Image to 3D Textured Mesh with Convolutional Reconstruction. ECCV 2024.
>
> [6] Meta 3D AssetGen: Text-to-Mesh Generation with High-Quality Geometry, Texture, and PBR Materials. NeurIPS 2024.
>
> [7] LRM-Zero: Training Large Reconstruction Models with Synthesized Data. NeurIPS 2024.
>
> [8] GeoLRM: Geometry-Aware Large Reconstruction Model for High-Quality 3D Gaussian Generation. NeurIPS 2024.
>
> [9] InstantMesh: Efficient 3D Mesh Generation from a Single Image with Sparse-View Large Reconstruction Models. arXiv 2024.
>
> [10] Tencent Hunyuan3D-1.0: A Unified Framework for Text-to-3D and Image-to-3D Generation. arXiv 2024.
>
> **W2-6: Many components are proposed before or only bring marginal improvements.**
>
> Regarding specific technical components, we highlight our key differences and improvements compared to previous work:
>
> FlexRM (Stage 2): FlexRM, unlike previous works [11, 12] that rely on a position prediction branch to determine 3D Gaussian positions, directly determines 3D Gaussian positions from the tri-plane representation using our proposed designs. Furthermore, FlexRM can process up to 32 input views and demonstrates significantly stronger performance than [11, 12].
>
> Multi-view image generation (Stage 1): Our contribution involves generating novel views with separate models for elevation and azimuth angles, minimizing multi-view conflicts and achieving better results compared to single-model approaches like SV3D [13] (We are happy to provide further visual results if necessary). Besides this, our proposed view curation pipeline effectively removes suboptimal or 3D-inconsistent generated views, improving the final 3D asset quality.
>
> Camera Conditioning (Stage 2): This minor design choice improves handling of multiple input views, especially since reconstruction models are trained with a varying number of them. As shown in lines 500-504 in the revised manuscript, the benefits of stronger camera conditioning become more effective with a larger number of input views. This modification is very simple and introduces negligible computational overhead.
>
> Imperfect Data Simulation (Stage 2): This novel contribution enhances FlexRM's robustness to minor imperfections in generated multi-view images, improving performance, particularly in generative tasks. While gains are marginal for reconstruction tasks (Table 5, right side), they are non-marginal for generation tasks (Table 5, left side), where the CLIP text similarity score increases from 27.1% to 27.7%. This metric is typically very close (<2% difference) among different methods.
>
> [11] Triplane Meets Gaussian Splatting: Fast and Generalizable Single-View 3D Reconstruction with Transformers, CVPR 2024.
>
> [12] AGG: Amortized Generative 3D Gaussians for Single Image to 3D, TMLR 2024
>
> [13] SV3D: Novel Multi-view Synthesis and 3D Generation from a Single Image using Latent Video Diffusion, ECCV 2024.

---

> ### Author Response · Authors · 2024-11-21
> **Thank you & responses (2)**
>
> **W7: No computational cost analysis.**
>
> We fully concur that reporting computational cost is essential. Thank you for raising this point! Our initial submission included runtime information for some pipeline stages. For instance, we have reported that view selection takes less than a second, and FlexRM generates 1 million Gaussian points in under 0.5 seconds and renders in real-time, comparable to current feed-forward reconstruction models.
>
> Generating 20 views using two diffusion models takes approximately one minute on a single H100 GPU. This speed is comparable to that of video-based multi-view diffusion models like SV3D. We have added a note about this in the revised paper. The revised paper now includes the runtime of the entire pipeline.
>
> **Q: Insights for follow-up research.**
>
> For follow-up research, the key insight is that we introduced two ways to handle imperfect multi-view synthesis results in a common two-stage 3D generation pipeline: view selection to reduce input errors, and noise simulation to train a robust reconstructor. Each individual component within Flex3D can be easily adopted by future works. For example, the minimalist design of the FlexRM architecture allows for easy implementation in frameworks like Instant-3D, and it can directly replace existing feed-forward reconstruction models.
>
> **Q: A much larger improvement in performance expected.**
>
> Our evaluation is quite extensive, encompassing both generation and reconstruction tasks. We believe our results on generation represent a significant improvement. We compared our proposed Flex3D with seven recent and strong baselines. In our user study, conducted in a fair environment with randomly selected samples for comparison, Flex3D outperformed all other methods, achieving at least a 92.5% win rate.
>
> **Ethics concerns on data.**
>
> We thank the reviewer for raising this important point. We understand these concerns and take ethics and IP rights extremely seriously.
>
> The data used in this paper was not obtained by scraping the internet. The data was purchased from a well-respected and widely-known vendor of 3D graphic assets. We acquired a data license that explicitly allows use of the models in machine-learning applications, including all applications in this paper. We follow the terms and conditions of this license scrupulously, also based on internal legal advice. The exact commercial details of the license are, as it may be expected, confidential.

---

> ### Comment · Reviewer_oAc3 · 2024-11-26
>
> I have taken my time to review the other reviewer's opinions, which I summarized at the end of this message. From what I can see, with the exception of Sr7a (and 9shi who I filter out as an outlier given the short review), the other reviewers are telling a similar story.
>
> I agree with their suggestions as well. Were these "tricks" applied to a number of existing techniques, and would carry consistent benefits, I would have been more keen to see the paper published. But as we stand, these are improvements to a closed system that will be complex/impossible to reproduce, especially as I don't see any indication of code, data, or trained model being released.
>
> Which leads me back to my original point... what is the academic going to take away from this paper? Therefore, unless another reviewer champions the paper, and convinces me that it will be a mistake to not have this paper published in its current form, I am very likely to **lower my score** to reject.
>
> ```
> pAxN (marginally below, confident)
> - core contribution is relatively straightforward
> - demonstrate the degree to which this idea directly and significantly improves the two-stage pipeline
>
> f8fj (marginally above, confident)
> - optimizing existing methods rather than introducing fundamentally novel concepts
> - questions remain regarding the scalability and broader applicability of the complex multi-stage pipeline
>
> zKWm (marginally below, certain)
> - you may apply the view selection to baseline methods to check whether there are consistent improvements
> - use the same selected multi-view images to evaluate different reconstruction model
>
> Sr7a (marginally above, certain)
> - (offers to raise conditional on 3D metrics and datasets)
>
> oAc3 (marginally below, confident)
> - myself
>
> 9shi (marginally above, certain)
> - review contains little insights
> ```

---

> > ### Author Response · Authors · 2024-11-26
> > **Thank you & responses (4)**
> >
> > **Q3: What is the academic going to take away from this paper?**
> >
> > In our response to reviewer PAxN (posted after your further comments), we have further highlighted the potential implications and insights of our work for future research. We are also posting it here as our response to this question.
> >
> > We expand the discussion here to include the following topics:
> >
> > **Feed-forward 3D generation:** We anticipate that two-stage 3D generation pipelines will remain popular in the future due to their many advantages. For example, they can easily adopt pre-trained diffusion models, and sparse-view inputs greatly simplify the reconstruction process, often leading to the best results. This line of research can draw many useful implications from our work, which makes the question we are addressing even more important.
> >
> > The key insight is that we introduced a series of methods to handle imperfect multi-view synthesis results in the common two-stage 3D generation pipeline. Our whole Flex3D pipeline introduces little computational cost but yields significant performance and robustness gains, and it could serve as a common design pipeline for future research in 3D generation. Additionally, all individual components proposed in this work can be easily adopted by future research in 3D generation to improve performance. Similarly, design ideas analogous to the Flex3D pipeline could be readily adopted for large 3D scene generation.
> >
> > **Feed-forward 4D generation:** Moreover, our work could be beneficial for 4D generation, which is an even more challenging task that faces similar limitations to two-stage 3D generation pipelines. Our pipeline could be directly extended to handle 4D object generation tasks. One could first generate 64 views (16 time dimensions * 4 multi-views) by fine-tuning video-based diffusion models, then slightly modify the view selection pipeline to keep only those views consistent across multiple views and time dimensions. Then, extend FlexRM from a tri-plane to a hex-plane or additionally learn time offsets to enable 4D representation. This should yield a strong method for 4D asset generation.
> >
> > **Leveraging 3D understanding for generation:** Keypoint matching techniques are used in this work to effectively mitigate multi-view inconsistencies. We hope this will also inspire the 3D generation community to incorporate advanced techniques from the rapidly evolving field of 3D understanding. Recent advances in deep learning have led to significant developments in matching, tracking, deep structure from motion, and scene reconstruction. These advancements offer the 3D generation community useful tools (such as pose estimation), pseudo-supervision signals (e.g., pseudo-depth supervision), and new model design ideas.
> >
> > We hope new information here provides further information to help you evaluate the contribution and quality of our work. Thank you once again for your detailed feedback and careful review!

---

> ### Author Response · Authors · 2024-11-26
> **Thank you & responses (3)**
>
> Many thanks for taking the time to review the other reviewers' comments and our rebuttal! We sincerely appreciate you providing further feedback, and we especially value your constructive criticism regarding the insights and broader implications of this paper. We also thank you for summarizing the other reviewers' perspectives.
>
> As no further comments regarding computational cost, performance, and ethical concerns regarding the data have been mentioned, we would like to assume that these concerns have been adequately addressed. Please feel free to raise any concerns in them if you feel our rebuttal did not sufficiently address them.
>
> Regarding your further comments, we would like to offer a response:
>
> **Q1: Were these "tricks" applied to a number of existing techniques, and would carry consistent benefits, I would have been more keen to see the paper published.**
>
> Thank you for your transparency. We believe that all proposed "tricks" or components in this paper **have been fully validated through extensive experiments and ablation studies.**
>
> Our experiments cover 3D generation and reconstruction tasks, and for each task, we have included more, or on-par, metrics, baselines, and test set sizes compared to previous research on 3D feed-forward generation models. This provides strong evidence to demonstrate the effectiveness of our proposed Flex3D pipeline and FlexRM model.
>
> Regarding the effectiveness of all components, we have included very detailed ablation studies. Their impact is further validated through rigorous ablations. For instance, we studied and reported the results of four design choices in FlexRM, three design choices in the candidate view curation and generation pipeline, and the noise injection pipeline, with detailed ablation results reported for each. During discussions with other reviewers, we also added many new experiments to further validate and support the effectiveness of our proposed components.
>
> Please feel free to leave any further comments if the effectiveness of any proposed component requires further clarification. We are fully committed to addressing them promptly.
>
> **Q2: I don't see any indication of code, data, or trained model being released.**
>
> The open-source plan is under discussion, and we are awaiting further guidance on this matter. We would like to clearly outline the potential outcomes to help you assess the impact of open-sourcing.
>
> The raw data will not be released as it was acquired for internal use only. Furthermore, the two fine-tuned multi-view diffusion models are also unlikely to be released. However, we may be able to open-source the weights of the reconstruction model (FlexRM), the code for running FlexRM, and the code for the view selection pipeline.
>
> We would also like to note that many advanced models in recent years, particularly generative models, are not open-sourced due to various complex reasons, and **often this decision is not made by the authors**. While authors may hope to use public training datasets and public models to build new techniques, this is **extremely difficult** due to various reasons. Open-sourcing should not be a factor in diminishing a paper's contribution if sufficient re-implementation details are provided. Otherwise, it is **very unfair** to researchers who cannot freely conduct open-sourcing, and this goes against ICLR's wishes, as it is a premier gathering of professionals dedicated to the advancement of artificial intelligence. Researchers who cannot freely open-source data and models should not be excluded.
>
> Excluding the fine-tuning details of the two multi-view diffusion models, **we have included very detailed implementation and training information to facilitate re-implementation**. This includes the proposed view selection, FlexRM, and noise injection methods. We provide re-implementation information comprehensively; for example, we have included the 3D Gaussian parameterization details.

---

> > ### Comment · Reviewer_oAc3 · 2024-11-29
> >
> > If these tricks' effectiveness were tested on multiple methods, and the code public, my rating would have been a weak accept.
> > But given this is not the case, I am afraid I will hold my (lowered) score.

---

> ### Author Response · Authors · 2024-11-29
> **Open-source**
>
> Many thanks for your willingness to engage in further discussion, and thank you again for your transparency! We also want to express our gratitude for your responsible reviewing and your contributions to fostering open-source development within the academic community.
>
> We do hope to open-source everything, ideally by early next year (Jan or early Feb). Following the double-blind review policy, we cannot disclose any author information at this time. However, we can state that all authors involved in this work are **active and well-engaged open-source contributors** in the research community, contributing both codes of research papers and open-source public packages/libraries. Furthermore, the first author of this work has **open-sourced all previous single-first-authored papers** and maintains good GitHub practices, replying to almost all issues. This can be easily verified once the author information is made public after the paper decision, so we are completely truthful here.
>
> Regarding the open-source plan, as stated before, it is still under discussion and we are waiting for further guidance. However, we are fairly confident (with >80% probability) that we will be able to **open-source the weights of the reconstruction model (FlexRM), the code for running FlexRM, the code for the noise simulation pipeline, and the code for the view selection pipeline**. Therefore, to reproduce all the performance results of Flex3D, the only missing components would be the two multi-view generation diffusion models. We should be able to provide an unofficial alternative using other multi-view diffusion models, such as SV3D, as a replacement. However, we must acknowledge that the performance in generation tasks might slightly decrease as a consequence.
>
> Although we cannot 100% guarantee the open-source plan at this moment, everything stated here is asserted with high confidence. We hope this addresses your concerns regarding open-sourcing!

---

> > ### Comment · Reviewer_oAc3 · 2024-11-30
> >
> > Sorry, I am looking for a guarantee – you could post it now.
> >
> > For example, CVPR allows *anonymous* github repositories containing the code. Even if the code does not run, as I am aware it may take time to adapt the code to open-source execution, I would consider that a sufficient token of good will. I have seen just too many "code coming soon" papers or incomplete code releases to accept anything less than that.

---

> > > ### Comment · Reviewer_pAxN · 2024-12-03
> > >
> > > I do not believe that open-sourcing is the only necessary means to validate the effectiveness of the proposed method in this paper. I do not recommend using the lack of open-sourcing as a reason to assign a "reject" rating.

---

> ### Author Response · Authors · 2024-11-29
> **Effectiveness were tested on multiple methods**
>
> Although the effectiveness of each trick has been fully validated in the current framework, we are exploring their generalization ability across additional frameworks.
>
> Specifically, we will test **(1) a stronger camera condition** and **(2) view selection** by applying them to a **variant of the Instant-3D reconstructor**. This variant will be trained using between 1 and 16 views as inputs, rather than a fixed four views. Training will begin soon, and we will post the results before the discussion period concludes.
>
> We will also test the proposed view selection pipeline when applied on top of SV3D's generated views, where we will report **(3) text/single-image to 3D generation results when applied to SV3D + FlexRM**.

---

> > ### Author Response · Authors · 2024-12-03
> > **Results for effectiveness were tested on multiple methods**
> >
> > We have obtained the results for the aforementioned experiments. The first two tests evaluate whether the proposed approaches can generalize across different reconstruction models (Instant3D's reconstructor), while the last one assesses their ability to generalize across different multi-view diffusion models (SV3D).
> >
> > For Instant3D's reconstructor, it was trained on 140,000 data samples for 30 epochs. The model is initialized from the pre-trained Stage (1) model in FlexRM, which uses NeRF as 3D represnetation. For SV3D, we utilized the SV3D_p variant.
> >
> > In summary, the results indicate a positive outcome, that is, **our proposed approaches generalize across different reconstruction models and multi-view diffusion models.** Full results are detailed below:
> >
> > **(1) Stronger camera condition**
> >
> > Here we report the averaged results of the 1-view, 4-view, 8-view, and 16-view testing settings.
> >
> > | Reconstruction model | PSNR&uarr; | SSIM&uarr; | LPIPS&darr; | CLIP image sim&uarr;|
> > |---|---|---|---|---|
> > | Instant3D | 22.56 | 0.796 | 0.112 | 0.776 |
> > | + stronger camera cond | 22.61 | 0.799 | 0.109 | 0.780 |
> >
> > The improvement trend is consistent with the results observed in FlexRM. Specifically, with more input views, the benefits of stronger camera conditioning become increasingly apparent.
> >
> > **(2) View selection**
> >
> > Here we report the text/single-image to 3D generation experiments using Emu-generated views and Instant3D's reconstructor to produce the final 3D assets.
> >
> > | Method| CLIP text similarity &uarr; | VideoCLIP text similarity &uarr; |
> > |---|---|---|
> > | No selection | 0.264| 0.248 |
> > | With selection | 0.273 | 0.253 |
> >
> > These results demonstrate that our view selection strategy is independent of the reconstruction model and can be applied broadly.
> >
> > **(3) Text/single-image to 3D generation results when applied to SV3D + FlexRM**
> >
> > In this experiment, we replaced the fine-tuned Emu model for candidate view generation with SV3D. Specifically, we generated 20 views at the same elevation/azimuth angles as those used with Emu. While this may not yield the optimal performance for SV3D, the goal here is to test generalization.
> >
> > | Method| CLIP text similarity &uarr; | VideoCLIP text similarity &uarr; |
> > |---|---|---|
> > | No selection | 0.263| 0.246 |
> > | With selection | 0.271 | 0.250 |
> >
> > These results demonstrate that our view selection strategy is also independent of the multi-view diffusion model and can be applied broadly.

---

> ### Author Response · Authors · 2024-11-30
> **Response regarding open-source**
>
> Thank you again for your willingness to engage in further discussion promptly.
>
> We have been **fully transparent** about our open-sourcing plan, outlining what may be open-sourced, our confidence levels, what cannot be open-sourced, and our proposed alternative. Given that this is an open review process, and even if reviewers have reservations about the authors' claims, we have no reason to make any false statements.
>
> Open-sourcing any material relevant to this work requires approval. We are currently in the process of acquiring approval for open-sourcing, and we are also exploring the possibility of sharing code confidentially in the coming days. If permitted, we will provide an anonymous link to the code for reviewers, ACs, and PCs only. Finally, we wish to clarify that attaching code is not a mandatory requirement for ICLR or CVPR; however, we are making every effort to accommodate your request in this regard.

---

> > ### Author Response · Authors · 2024-12-03
> > **Update on code sharing**
> >
> > Authors are still waiting to hear back for further guidance to see if we are allowed to share code confidentially. We are actively following up to expedite the process.

---

> > > ### Author Response · Authors · 2024-12-04
> > > **Further update on code sharing**
> > >
> > > We have made every possible effort to obtain permission for confidential sharing (we initiated the request immediately after reviewer oAc3's request and have followed up three times). However, approval has not yet been granted due to the limited time available. We will provide updates as new developments arise. **If we are unable to respond to the reviewers after the discussion period ends, we will send a message to the ACs, provided that the OpenReview system allows it.**
> > >
> > > Our commitment to the previously shared open-sourcing plan remains unchanged, and we will continue to actively pursue open-sourcing this work.

---

### Official Review · Reviewer_Sr7a · 2024-11-04

**Soundness:** 2
**Presentation:** 3
**Contribution:** 2
**Rating:** 6
**Confidence:** 5

**Summary:**

Flex3D is a novel two-stage framework for generating high-quality 3D content from text prompts, single images, or sparse-view images. In the first stage, it generates a diverse pool of candidate views using fine-tuned multi-view image diffusion models and video diffusion models. A view selection pipeline then filters these views based on quality and consistency. The second stage employs the FlexRM, a transformer-based architecture capable of processing an arbitrary number of input views with varying viewpoints. FlexRM combines tri-plane features with 3D Gaussian Splatting to produce detailed 3D Gaussian representations. Experimental results demonstrate that Flex3D outperforms state-of-the-art methods in both 3D reconstruction and generation tasks.

**Strengths:**

The paper is well-organized, with a clear delineation of the contributions and methodologies. The progression from problem identification to solution proposal is logical and easy to follow.

The key contributions of this paper are two-fold, which seem to be of effectiveness according to the experimental analysis:
1. candidate view generation and curation: Introduction of a multi-view generation strategy that produces a diverse set of candidate views from varying azimuth and elevation angles, followed by a selection process that filters views based on quality and consistency.
2. flexible reconstruction model (FlexRM): Development of a robust 3D reconstruction network capable of ingesting an arbitrary number of input views with varying viewpoints. FlexRM efficiently processes these views to output high-quality 3D Gaussian representations using a combination of tri-plane features and 3D Gaussian Splatting.

The authors conduct detailed ablation studies to validate the effectiveness of each component of their proposed framework.

**Weaknesses:**

My major concerns lay in the following several aspects. If some of the may concerns can be solved during the discussion section, I would like to raise the final score.

1. The paper does not specify whether the proposed method has been tested across various datasets or object categories. Evaluating Flex3D on diverse and challenging datasets would demonstrate its generalizability and robustness to different types of input data.

2. The paper evaluates performance using 2D image-based metrics such as PSNR, SSIM, LPIPS, and CLIP image similarity. While these metrics are informative, they do not fully capture the geometric accuracy and consistency of the 3D reconstructions. Incorporating 3D-specific metrics, such as Chamfer Distance or Earth Mover's Distance, would provide a more comprehensive assessment of the reconstructed 3D models' quality.

3. The user study conducted to evaluate the overall quality of the generated content lacks detailed methodology. Information regarding participant demographics, selection criteria, and statistical significance testing is absent. Providing these details would enhance the credibility of the user study findings.

**Questions:**

See the weakness section.

---

> ### Author Response · Authors · 2024-11-21
> **Thank you & responses (1)**
>
> **W1: The paper does not specify whether the proposed method has been tested across various datasets or object categories.**
>
> Flex3D primarily focuses on text- or single-image-based 3D generation, while FlexRM supports various 3D reconstruction tasks. For generation, we used a diverse set of 404 DreamFusion prompts encompassing a wide range of objects, scenes, styles, and abstract descriptions—a relatively large experiment compared to many prior works. For reconstruction, we tested on 947 real-world scanned objects from the GSO dataset, covering common object categories (excluding a few highly similar shoes to avoid redundancy). This test set is also larger than those used in many previous works, which often employ fewer than a few hundred GSO samples.
>
> To further validate FlexRM's robustness across diverse data distributions, we tested it on Blender-rendered views of 500 hand-crafted 3D objects from our internal dataset (similar to Objaverse). This validation set was held out from training. Similar to the GSO procedure, we rendered 64 views per object at four elevation degrees (-30, 6, 30, and 42 degrees), with 16 uniformly distributed azimuth angles per elevation. Results are presented in Table 6 (page 18) of the revised manuscript, showing similar trends to the GSO results.
>
> Finally, we tested FlexRM's robustness to noisy input images as a more diverse case. The results are presented in our response to reviewer f8fj (Q1: Results on robustness across various poses, view counts, and noise levels.).
>
> **W2: Adding these 3D metrics would provide a more complete understanding of the method's performance.**
>
> We agree that incorporating more 3D metrics leads to a more complete evaluation—thank you for highlighting this! We further report the Chamfer Distance and Normal Correctness for 20,000 points uniformly sampled on both the predicted and ground-truth shapes in the GSO experiment. Results are presented in Table 2 of the revised manuscript. FlexRM continues to outperform other baselines in the 3D evaluation metrics, by clear margins, and we also observe improved results with more input views.
>
> **W3: Providing these details would enhance the credibility of the user study findings.**
>
> We provide additional details about the user study described in lines 443-450 of the revised manuscript. Our study design generally follows previous work [1-3].
>
> **Participant Demographics**: Five computer vision or machine learning researchers participated in the evaluation. Two were from the US, two from Europe, and one from Asia. Two participants actively work on 3D content generation.
>
> **Methodology**: Participants viewed paired 360° rendered videos—one generated by Flex3D and one by a baseline method—presented via a Google Form. Video pairs were presented in random order and randomized left/right positions. Participants selected the video they preferred based on overall visual quality.
>
> **Statistical Significance**: We collected 1000 valid results (5 participants * 5 baselines * 40 videos). Although the sample size is relatively small, Flex3D was preferred in at least 92.5% of comparisons across all five baselines, strongly suggesting superior visual quality.
>
> **Update**: During the rebuttal period, we added comparisons with two direct 3D diffusion baselines using the same user study setup, bringing the total number of valid results to 1400.
>
> [1]: IM-3D: Iterative Multiview Diffusion and Reconstruction for High-Quality 3D Generation, ICML 2024.
>
> [2]: Emu Video: Factorizing Text-to-Video Generation by Explicit Image Conditioning, ECCV 2024.
>
> [3]: VFusion3D: Learning Scalable 3D Generative Models from Video Diffusion Models,  ECCV 2024.

---

> > ### Author Response · Authors · 2024-11-28
> > **Gentle reminder**
> >
> > Dear Reviewer Sr7a:
> >
> > We sincerely appreciate the time and effort you dedicated to reviewing our paper! In response to your concerns, we have conducted additional experiments on more datasets and reported 3D metrics during the discussion period.
> >
> > As the discussion period concludes soon, we kindly request, if possible, that you review our rebuttal at your convenience. Should there be any further points requiring clarification or improvement, we are fully committed to addressing them promptly. Thank you once again for your invaluable contribution to our manuscript!
> >
> > Warm regards,
> > The Authors

---

### Official Review · Reviewer_zKWm · 2024-11-04

**Soundness:** 3
**Presentation:** 4
**Contribution:** 2
**Rating:** 5
**Confidence:** 5

**Summary:**

This work focuses on feed-forward 3d generation. Following previous work, this paper adopts a synthesis-then-reconstruction method, where a multi-view diffusion generates multiple images at different camera views, and a regression model then reconstructs 3d representation based on multi-view images. The main contribution the author claimed is the view selection trick that curates generated multi-view images based on the back-view quality and consistency. Also, the proposed method uses 3DGS as a 3d representation for rendering efficiency.

**Strengths:**

- The writing is well-organized and easy to follow.
- This work proposed to select condition images from the generated multi-view images based on the quality, thereby improving the 3d reconstruction quality.

**Weaknesses:**

- This work basically follows previous work like Instant3D and replaces the triplane NeRF representation with triplane Gaussian (as in [1]). The main contribution thus lies in the candidate view selection. It is evident that the reconstruction quality would improve with better generated multi-view images, but the key is how to define 'better' and automatically filter the better images. The proposed method adopts SVM trained with 2,000 manually labeled data to select back view, but the paper does not describe how to label the data and does not give the criterion. Also, 2,000 images are small and restricted by the bias of labelers. This would lead to very biased and uninterpretable labels for training a good classifier. How about the success rate of the selection model? How to determine whether it is a good classification? There is a lack of sufficient analysis and experiments that support the claim. There are similar concerns to the consistency selection model. Why do you choose manually crafted rules for selection, like using DINO, SVM, LOFTER? Are they the best choices? Any insights?
- Based on the aforementioned comment, I would suggest the authors to compare with automatic selection with large multimodal model like GPT4V. It is straightforward to give the grid of images to the large model, and ask it to select images. Would it be better than the proposed method?
- There is a lack of comparison with diffusion-based baselines that predict 3d via 3d diffusion or 3d dit directly.
- The proposed method comprises two stages. Which stage does the improvement mainly come from? multi-view generation and selection, or flex reconstruction model? In Fig.4, and table 1, do the baselines use the same multi-view images as the proposed method? I would suggest evaluating two stages separately. Specifically, you may apply the view selection to baseline methods to check whether there are consistent improvements. Also, use the same selected multi-view images to evaluate different reconstruction model.
- For ablation results like table 3,4,5, do you use Blender rendered images or generated images as the multi-view condition? Could the data simulation address the domain gap of data? How about metrics in table 5 using GT multi-view images rather than generated multi-view images?


[1] Triplane Meets Gaussian Splatting: Fast and Generalizable Single-View 3D Reconstruction with Transformers

**Questions:**

How many views are used for calculation metrics for Flex3d in Table 1? More than baseline methods? If so, is the comparison fair?

---

> ### Author Response · Authors · 2024-11-21
> **Thank you & responses (1)**
>
> **W1: Lacks sufficient detail and analysis regarding the data labeling process, classifier performance, and justification for the chosen selection strategies.**
>
> Regarding video quality selection criteria, our Emu-generated videos contain 16 frames each. Our criteria are based on the overall visual quality of the generated multi-view videos, emphasizing multi-view consistency and visual appearance. For labeling, the authors first carefully labeled approximately 100 sample videos. These were then provided to two labelers, and each labeler was asked to label approximately 1,000 videos, resulting in a total of 2,000 labeled videos. A sample size of 2,000 might be considered small. This number follows the setting used in Instant-3D's data curation pipeline. Similar to Instant-3D, we find that using strong pre-trained image feature encoders like DINO allows 2,000 samples to be sufficient for training a robust classifier using SVM. Other pre-trained image feature extractors, such as CLIP, can also be effective, but DINO proved more effective in our experiments. We found that SVM converges more easily than neural net-based linear classifiers. To further verify the success rate, we manually labeled another 100 videos and applied the trained classifier. It achieved aligned results for 93 videos, which we consider a high success rate.
>
> For the consistency selection model, we chose EfficientLoFTR [1] because it represents the state-of-the-art in real-time keypoint matching. In multi-view geometry, 3D consistency primarily relies on 2D correspondences established through keypoint matching. Over the past few decades, keypoint matching methods have evolved from classical approaches like SIFT combined with nearest neighbor search to modern deep learning techniques such as SuperPoint [2] with SuperGlue [3], and EfficientLoFTR. These methods have proven reliable and effective and are extensively used in 3D reconstruction tasks like structure from motion. In our evaluations, EfficientLoFTR performed best, producing all results in under a second. While other viable options exist, such as SuperPoint with SuperGlue, they often do not offer the same optimal balance of speed and accuracy.
>
> [1] Efficient LoFTR: Semi-dense local feature matching with sparse-like speed. CVPR 2024.
>
> [2] Superpoint: Self-supervised interest point detection and description. CVPRW 2018.
>
> [3] Superglue: Learning feature matching with graph neural networks. CVPR 2020.
>
> [4] LoFTR: Detector-free local feature matching with transformers. CVPR 2021.
>
> **W2:  A comparison with selection via MLLMs like GPT-4V.**
>
> This is a great idea; thank you for suggesting it! We tested it using Gemini 1.5 Pro 002, which may be more powerful than GPT4V due to its long context window, allowing us to input all 20 frames directly. Our prompt was: “You are an expert in 3D computer vision. I am providing you with 20 generated multi-view images. Please help me identify the frames that are consistent with the first frame and present high visual quality. Don't be too strict, as minor inconsistencies are acceptable.”  This resulted in a total token count of 5,224, much smaller than Gemini 1.5 Pro 002's context window (2M).
>
> We tested 20 samples. Generally, Gemini can understand the task and make reasonable selections, but its performance is not yet as good as our proposed pipeline. This might be because current MLLMs primarily rely on CLIP embeddings to connect visual modalities, making pixel-level perception difficult.
>
> We describe two specific selection results from two sequences used in Figure 5, showing all 20 frames. For the "ramen" example, Gemini selected frames [1, 5, 7, 13, 17], while our pipeline selected [1, 2, 3, 5, 6, 10, 11, 12, 13, 18, 19, 20]. Compared with our pipeline, Gemini rejected frames [2, 3, 6, 10, 11, 12, 18, 19, 20] and selected [7, 17]. Upon careful inspection, frames [2, 3, 6, 10, 11, 12, 19, 20] should be considered high-quality and retained. In Gemini's selected frames [7, 17], the chopsticks are either missing or blurry and should be removed.
>
> For the "robot and dragon playing chess" example, Gemini selected frames [1, 2, 3, 5, 6, 8, 9, 11, 12, 14, 15, 17, 18, 20], while our pipeline selected [1, 2, 3, 4, 5, 6, 12, 13, 14, 20]. Compared with our pipeline, Gemini rejected frames [4, 13] and selected [8, 9, 11, 15, 17, 18]. Frames [4, 13] should be considered high-quality and retained. In Gemini's selected frames [8, 9, 11], the robot's eyes are incorrect, either missing or merged. In frames [15, 17, 18], the dragon's head is incorrect, displaying an unusual shape or blue color.
>
> In conclusion, while MLLMs are not yet as effective or efficient as our proposed pipeline (where our proposed pipeline also runs faster, in under a second), using them for view selection holds strong potential. Future research could address these limitations through fine-tuning to improve pixel-level perception and 3D awareness. We leave this interesting idea for future work.

---

> ### Author Response · Authors · 2024-11-21
> **Thank you & responses (2)**
>
> **W3: There is a lack of comparison with diffusion-based baselines that predict 3d via 3d diffusion or 3d dit directly.**
>
> We've added results from two recent, open-source, direct 3D diffusion models for comparison: LN3Diff [1] and 3DTopia-XL [2]. Using their official code, we generated 3D assets via their single-image-to-3D pipelines. This allows more controlled generation, and was necessary because the text-to-3D pipeline is not currently available for 3DTopia-XL. We've updated Section 4.1 on 3D generation, including Figure 4 and Table 1. Overall, Flex3D significantly outperforms these direct 3D baselines, achieving considerably higher CLIP scores and demonstrating a 95% and 97.5% win rate in user studies.
>
> [1] LN3Diff: Scalable Latent Neural Fields Diffusion for Speedy 3D Generation, ECCV 2024.
>
> [2] 3DTopia-XL: High-Quality 3D PBR Asset Generation via Primitive Diffusion, arXiv 2024.
>
> **W4: Which stage does the improvement mainly come from?**
>
> We would state the improvement comes slightly more from the multi-view generation and selection stage. However, to utilize the selected multi-views with a varying number for reconstruction, a reconstruction model capable of handling any number of input views (like FlexRM) is necessary.
>
> **W4: In Fig.4, and table 1, do the baselines use the same multi-view images as the proposed method?**
>
> No, Figure 4 and Table 1 present the results of text-to-3D generation tasks. The input for each method is a text prompt or a single image generated from a text prompt.
>
> **W4: Evaluating two stages separately:**
>
> We agree that isolating the contributions of the multi-view generation and selection stage versus the FlexRM reconstruction model would provide valuable insights.
>
> **Multi-view generation and selection**: Since other reconstruction models are trained with a fixed number of views (e.g., LGM-4, GRM-4, InstantMesh-6, Instant3D-4, LRM-1, VFusion3D-1), directly adapting them to variable view settings leads to performance degradation, hindering a clear comparison of our view selection strategy. Therefore, we included a detailed ablation study (Table 4) to demonstrate the effects of our proposed multi-view generation and selection pipeline.
>
> **FlexRM reconstructor**: The reconstruction task (where the input views are identical), presented in Section 4.2, shows detailed reconstruction results in different settings with comparisons to representative baselines. FlexRM consistently outperforms other baselines, achieving the best results across different input view settings.
>
> **W5: For ablation results like table 3,4,5, do you use Blender rendered images or generated images as the multi-view condition?**
>
> We used two distinct settings for ablations: one for reconstruction (Table 3 and the right side of Table 5) and one for generation (Table 4 and the left side of Table 5). For reconstruction, we rendered GSO scanned objects in Blender. For generation, we used generated images and text-prompt as the condition.
>
> **W5: How about metrics in table 5 using GT multi-view images rather than generated multi-view images?**
>
> Concerning the metrics in Table 5, the right side (reconstruction results) utilizes GT multi-view images for evaluation. The reported results are averaged across experiments using 1, 4, 8, and 16 input views using GSO (scanned objects) as benchmark. Table 5 demonstrates that our data simulation leads to a reasonable performance improvement in generative tasks and a marginal improvement in reconstruction tasks.
>
> **W5: Could the data simulation address the domain gap of data?**
>
> While the primary purpose of the data simulation is to enhance FlexRM's robustness to minor imperfections in generated input multi-view images, thereby improving its performance in generation tasks, it also indirectly addresses the domain gap. By increasing the diversity of the input data during training, the data simulation can slightly mitigate domain gap issues. However, we have to state that it is not a primary solution for large domain gaps and using it alone won't fully resolve such issues.
>
> **Q1: How many views are used for calculation metrics for Flex3d in Table 1? More than baseline methods? If so, is the comparison fair?**
>
> Table 1 presents results for text-to-3D generation tasks, where the input for each method is a text prompt or a single image generated from a text prompt. The number of views used for metric calculation (CLIP and Video-Clip)  is identical (240) for all methods’ rendered video results. The comparison is consistent with many established practices in this field and is fair.

---

> ### Comment · Reviewer_zKWm · 2024-11-27
> **Respose to authros**
>
> Thanks for your responses to my questions. My concerns about the implementation details have been addressed. However, I agree with other reviewers that this paper does not bring many new things to the community. So, I would keep my scores for the current version. Thanks for your efforts and active responses. Good luck!

---

> > ### Author Response · Authors · 2024-11-27
> >
> > We sincerely appreciate the time and effort you dedicated to providing detailed and thoughtful feedback on our manuscript! We are grateful for your careful reconsideration of our paper, our rebuttal, and the other reviews. We are particularly intrigued by your suggestion of exploring the use of MLLMs for view selection and thank you for proposing such an interesting idea!
> >
> > We are pleased that the rebuttal has satisfactorily addressed all concerns regarding implementation details.
> >
> > We understand your new concerns, as well as those of the other reviewers, regarding insights and implications for future work. We have tried our best to address this, and we now present a discussion of how our work can be useful for directions such as feed-forward 3D generation, feed-forward 4D generation, and leveraging 3D understanding for generation. In the discussion, we have also outlined a concrete future research idea on 4D generation. Please feel free to review these additions (attached below) if you have not already done so.
> >
> > Thank you once again for your invaluable feedback. It was truly a pleasure to have you as the reviewer, and your suggestions have definitely strengthened our paper!
> >
> > **Discussion:**
> >
> > **Feed-forward 3D generation:** We anticipate that two-stage 3D generation pipelines will remain popular in the future due to their many advantages. For example, they can easily adopt pre-trained diffusion models, and sparse-view inputs greatly simplify the reconstruction process, often leading to the best results. This line of research can draw many useful implications from our work, which makes the question we are addressing even more important.
> >
> > The key insight is that we introduced a series of methods to handle imperfect multi-view synthesis results in the common two-stage 3D generation pipeline. Our whole Flex3D pipeline introduces little computational cost but yields significant performance and robustness gains, and it could serve as a common design pipeline for future research in 3D generation. Additionally, all individual components proposed in this work can be easily adopted by future research in 3D generation to improve performance. Similarly, design ideas analogous to the Flex3D pipeline could be readily adopted for large 3D scene generation.
> >
> > **Feed-forward 4D generation:** Moreover, our work could be beneficial for 4D generation, which is an even more challenging task that faces similar limitations to two-stage 3D generation pipelines. Our pipeline could be directly extended to handle 4D object generation tasks. One could first generate 64 views (16 time dimensions * 4 multi-views) by fine-tuning video-based diffusion models, then slightly modify the view selection pipeline to keep only those views consistent across multiple views and time dimensions. Then, extend FlexRM from a tri-plane to a hex-plane or additionally learn time offsets to enable 4D representation. This should yield a strong method for 4D asset generation.
> >
> > **Leveraging 3D understanding for generation:** Keypoint matching techniques are used in this work to effectively mitigate multi-view inconsistencies. We hope this will also inspire the 3D generation community to incorporate advanced techniques from the rapidly evolving field of 3D understanding. Recent advances in deep learning have led to significant developments in matching, tracking, deep structure from motion, and scene reconstruction. These advancements offer the 3D generation community useful tools (such as pose estimation), pseudo-supervision signals (e.g., pseudo-depth supervision), and new model design ideas.

---

### Author Response · Authors · 2024-11-21
**General response**

Thank you to all six reviewers for your thoughtful and detailed feedback on Flex3D! We are encouraged that many reviewers found our submission to be **well-written, organized, and clearly delineated (zKWm, Sr7a, f8fj, 9shi)**. We appreciate reviewers’ recognition of our approach as **well-motivated (f8fj, pAxN) and novel (9shi, pAxN)**. We are also pleased that reviewers found our **visual results compelling (oAc3, 9shi)** and our **experiments solid and comprehensive (Sr7a, pAxN)**.

We will address individual questions in separate threads. Thank you again for your time and further discussion! We are happy to answer any follow-up questions.

---

> ### Author Response · Authors · 2024-11-25
> **Gentle reminder**
>
> Dear Reviewers:
>
> Once again, we sincerely appreciate the time and effort you have dedicated to reviewing our paper!
>
> As the discussion period concludes in two days, we would be grateful if you could review our rebuttal at your convenience, should your schedule allow. If there are any further points requiring clarification or improvement, please be assured that we are fully committed to addressing them promptly. Thank you once again for your invaluable feedbacks to our research!
>
> Warm regards,
> The Authors

---

### Author Response · Authors · 2024-12-04
**Updated version for future research insights**

We would like to share an updated version focusing on the primary concern of this work: how it can inspire future research. The key difference is the inclusion of a new research area, **Generative 3D/4D reconstruction**, which could also benefit from our work.

**Generative 3D/4D reconstruction**:
Methods like Im-3D, Cat3D, and Cat4D rely on multi-view diffusion models to synthesize a large number of possible views, which are then used to fit a 3D/4D representation. Our work could inspire advancements in this area in two key ways:

View Selection: Our view selection pipeline could directly enhance these methods by filtering out inconsistent synthesized views before fitting a 3D representation, potentially leading to performance improvements. This approach can also be naturally extended to handle synthesized 4D views.

Efficiency: Fitting a 3D representation from scratch is time-consuming. The FlexRM reconstruction model we developed can process up to 32 views and has the potential to scale further. Synthesized views could first be processed through this model, which operates in under a second, to generate a strong initial 3D representation. This approach has the potential to dramatically reduce the time required for fitting a 3D representation—from the usual half an hour to under a minute. This idea could also be extended to 4D, as adapting the current 3D reconstruction pipeline to 4D is straightforward.

**Feed-forward 3D generation**: We anticipate that two-stage 3D generation pipelines will remain popular in the future due to their many advantages. For example, they can easily adopt pre-trained diffusion models, and sparse-view inputs greatly simplify the reconstruction process, often leading to the best results. This line of research can draw many useful implications from our work, which makes the question we are addressing even more important.

The key insight is that we introduced a series of methods to handle imperfect multi-view synthesis results in the common two-stage 3D generation pipeline. Our whole Flex3D pipeline introduces little computational cost but yields significant performance and robustness gains, and it could serve as a common design pipeline for future research in 3D generation. Additionally, all individual components proposed in this work can be easily adopted by future research in 3D generation to improve performance. Similarly, design ideas analogous to the Flex3D pipeline could be readily adopted for large 3D scene generation.

**Feed-forward 4D generation**: Moreover, our work could be beneficial for 4D generation, which is an even more challenging task that faces similar limitations to two-stage 3D generation pipelines. Our pipeline could be directly extended to handle 4D object generation tasks. One could first generate 64 views (16 time dimensions * 4 multi-views) by fine-tuning video-based diffusion models, then slightly modify the view selection pipeline to keep only those views consistent across multiple views and time dimensions. Then, extend FlexRM from a tri-plane to a hex-plane or additionally learn time offsets to enable 4D representation. This should yield a strong method for 4D asset generation.

**Leveraging 3D understanding for generation**: Keypoint matching techniques are used in this work to effectively mitigate multi-view inconsistencies. We hope this will also inspire the 3D generation community to incorporate advanced techniques from the rapidly evolving field of 3D understanding. Recent advances in deep learning have led to significant developments in matching, tracking, deep structure from motion, and scene reconstruction. These advancements offer the 3D generation community useful tools (such as pose estimation), pseudo-supervision signals (e.g., pseudo-depth supervision), and new model design ideas.

---

### Meta-Review · Area_Chair_yFRY · 2024-12-20

**Metareview:**

In this paper, the authors have proposed Flex3D which is a feedforward method by first generating multi-view images and then reconstructing 3D contents. In the first stage, it selects views according to the quality and consistency. In the second stage, FlexRM is proposed to reconstruct 3D Gaussians. The paper obtains mixed scores of 5 and 6. From my point of view, the key limitation of this paper is the technical contributions are too engineering without providing more scientific insights, which is very important for a top conference paper. Also, the experimental improvements are not significant. Due to the reasons especially on the scientific novelty, I recommend a decision of rejection.

**Additional Comments On Reviewer Discussion:**

Initially the reviewers raised concerns on the novelty, technical contributions, the pipeline, experimental comparisons, metrics, ablation studies, and missing references. The authors have addressed some of them, but due to the limited novelty and technical contributions, I still recommend a decision of rejection.

---

### Decision · Program_Chairs · 2025-01-22

Reject